# Multi-Agent Reinforcement Learning in Stochastic Networked Systems

**Yiheng Lin**
CMS, Caltech
yihengl@caltech.edu

**Guannan Qu**
CMS, Caltech
gqu@caltech.edu

**Longbo Huang**
IIIS, Tsinghua University
longbohuang@tsinghua.edu.cn

**Adam Wierman**
CMS, Caltech
adamw@caltech.edu

## Abstract

We study multi-agent reinforcement learning (MARL) in a stochastic network of agents. The objective is to find localized policies that maximize the (discounted) global reward. In general, scalability is a challenge in this setting because the size of the global state/action space can be exponential in the number of agents. Scalable algorithms are only known in cases where dependencies are static, fixed and local, e.g., between neighbors in a fixed, time-invariant underlying graph. In this work, we propose a Scalable Actor Critic framework that applies in settings where the dependencies can be non-local and stochastic, and provide a finite-time error bound that shows how the convergence rate depends on the speed of information spread in the network. Additionally, as a byproduct of our analysis, we obtain novel finite-time convergence results for a general stochastic approximation scheme and for temporal difference learning with state aggregation, which apply beyond the setting of MARL in networked systems.

## 1 Introduction

Multi-Agent Reinforcement Learning (MARL) has achieved impressive performance in a wide array of applications including multi-player game play [42, 31], multi-robot systems [13], and autonomous driving [25]. In comparison to single-agent reinforcement learning (RL), MARL poses many challenges, chief of which is scalability [57]. Even if each agent's local state/action spaces are small, the size of the global state/action space can be large, potentially exponentially large in the number of agents, which renders many RL algorithms such as $Q$-learning not applicable.

A promising approach for addressing the scalability challenge that has received attention in recent years is to exploit application-specific structures, e.g., [18, 35, 38]. A particularly important example of such a structure is a networked structure, e.g., applications in multi-agent networked systems such as social networks [7, 27], communication networks [60, 51], queueing networks [34], and smart transportation networks [59]. In these networked systems, it is often possible to exploit *static, local* dependency structures [16, 17, 1, 32], e.g., the fact that agents only interact with a fixed set of neighboring agents throughout the game. This sort of dependency structure often leads to scalable, distributed algorithms for optimization and control [16, 1, 32], and has proven effective for designing scalable and distributed MARL algorithms, e.g. [35, 38].

This work was supported by NSF grants CNS-2106403 and NGSDI-2105648, with additional support from Amazon AWS, PIMCO, and the Resnick Sustainability Insitute. Yiheng Lin was supported by Kortschak Scholars program. The work of Longbo Huang was supported by the Technology and Innovation Major Project of the Ministry of Science and Technology of China under Grants 2020AAA0108400 and 2020AAA0108403.

35th Conference on Neural Information Processing Systems (NeurIPS 2021).

However, many real-world networked systems have inherently *time-varying, non-local* dependencies. For example, in the context of wireless networks, each node can send packets to other nodes within a fixed transmission range. However, the interference range, in which other nodes can interfere the transmission, can be larger than the transmission range [53]. As a result, due to potential collisions, the local reward of each node not only depends on its own local state/action, but also depends on the actions of other nodes within the interference range, which may be more than one-hop away. In addition, a node may be able to observe other nodes' local states before picking its local action [33]. Things become even more complex when mobility and stochastic network conditions are considered. These lead to dependencies that are both stochastic and non-local. Although one can always fix and localize the dependence model, this leads to considerably reduced performance. Beyond wireless networks, similar stochastic and non-local dependencies exists in epidemics [30], social networks [7, 27], and smart transportation networks [59].

A challenging open question in MARL is to understand how to obtain algorithms that are scalable in settings where the dependencies are stochastic and non-local. Prior work considers exclusively static and local dependencies, e.g., [35, 38]. It is clear that hardness results apply when the dependencies are too general [24]. Further, results in the static, local setting to this point rely on the concept of exponential decay [35, 16], meaning the agents' impact on each other decays exponentially in their graph distance. This property relies on the fact that the dependencies are purely local and static, and it is not clear whether it can still be exploited when the interactions are more general. This motivates an important open question: *Is it possible to design scalable algorithms for stochastic, non-local networked MARL?*

**Contributions.** In this paper, we introduce a class of stochastic, non-local dependency structures where every agent is allowed to depend on a random subset of agents. In this context, we propose and analyze a Scalable Actor Critic (SAC) algorithm that provably learns a near-optimal local policy in a scalable manner (Theorem 2.5). This result represents the *first* provably scalable method for stochastic networked MARL. Key to our approach is that the class of dependencies we consider leads to a $\mu$-decay property (Definition 2.1). This property generalizes the exponential decay property underlying recent results such as [35, 16], which does not apply to stochastic non-local dependencies, and enables the design of an efficient and scalable algorithm for settings with stochastic, non-local dependencies. Our analysis of the algorithm reveals an important trade-off: as deeper interactions appear more frequently, the "information" can spread more quickly from one part of the network to another, which leads to the efficiency of the proposed method to degrade. This is to be expected, as when the agents are allowed to interact globally, the problem becomes a single-agent tabular $Q$-learning problem with an exponentially large state space, which is known to be intractable since the sample complexity is polynomial in the size of the state/action space [12, 24].

The key technical result underlying our analysis of the Scalable Actor Critic algorithm is a finite-time analysis of a general stochastic approximation scheme featuring infinite-norm contraction and state aggregation (Theorem 3.1). We apply this result to networked MARL using the local neighborhood of each agent to provide state aggregation (SA). This result also applies beyond MARL. Specifically, we show that it yields finite-time bounds on Temporal Difference (TD)/$Q$ learning with state aggregation (Theorem 3.2). To the best of our knowledge the resulting bound is the first finite-time bound on asynchronous $Q$-learning with state aggregation. Additionally, it yields a novel analysis for TD-learning with state aggregation (the first error bound in the infinity norm) that sheds new insight into how the error depends on the quality of state abstraction. These two results are important contributions in their own right. Due to space constraints, we discuss asynchronous $Q$-learning with state aggregation in Appendix D.4.

**Related literature.** The prior work that is most related to our paper is [38], which also studies MARL in a networked setting. The key difference is that we allow the dependency structure among agents to be non-local and stochastic, while [38] requires the dependency structure to be local and static. The generality of setting means techniques from [38] do not apply and adds considerable complexity to the proof in two aspects. First, instead of analyzing the algorithm directly like [38], we derive a finite-time error bound for TD learning with state aggregation (Section 3.1 and 3.2), and then establish its connection with the algorithm (Section 2.3). Second, we need a more general decay property (Definition 2.1) than the exponential one used in [38]. Defining and establishing this general decay property for the non-local and stochastic setting is highly non-trivial (Section 2.1).

More broadly, MARL has received considerable attention in recent years, see [57] for a survey. The line of work most relevant to the current paper focuses on cooperative MARL. In the cooperative setting, each agent can decide its local actions but share a common global state with other agents. The objective is to maximize a global reward by working cooperatively. Notable examples of this approach include [6, 10] and the references therein. In contrast, we study a situation where each agent has its own state that it acts upon. Despite the differences, like our situation, cooperative MARL problems still face scalability issues since the joint-action space is exponentially large. A variety of methods have been proposed to deal with this, including independent learners [8, 29], where each agent employs a single-agent RL policy. Function approximation is another approach that can significantly reduce the space/computational complexity. One can use linear functions [58] or neural networks [28] in the approximation. A limitation of these approaches is the lack of theoretical guarantees on the approximation error. In contrast, our technique not only reduces the space/computational complexity significantly, but also has theoretical guarantees on the performance loss in settings with stochastic and non-local dependencies.

The mean-field approach [45, 56, 19] provides another way to address the scalability issue, but under very different settings compared to ours. Specifically, the mean-field approach typically assumes homogeneous agents with identical local state/action space and policies, and each agent depends on other agents through their population or "mean" behavior. In contrast, our approach considers a local-interaction model, where there is an underlying graph and each agent depends on neighboring agents in the graph. Further, our approach allows heterogeneous agents, which means that the local state/action spaces and policies can differ among the agents.

Another related line of work uses centralized training with decentralized execution, e.g., [28, 15], where there is a centralized coordinator that can communicate with all the agents and keep track of their experiences and policies. In contrast, our work only requires distributed training, where we constrain the scale of communication in training within the $\kappa$-hop neighborhood of each agent.

More broadly, this paper contributes to a growing literature that uses exponential decay to derive scalable algorithms for learning in networked systems. The specific form of exponential decay that we generalize is related to the idea of "correlation decay" studied in [16, 17], though their focus is on solving static combinatorial optimization problems whereas ours is on learning policies in dynamic environments. Most related to the current paper is [38], which shows an exponential decay property in a restricted networked MARL model with purely local dependencies. In contrast, we show a more general $\mu$-decay property holds for a general form of stochastic, non-local dependencies.

The technical work in this paper contributes to the analysis of stochastic approximation (SA), which has received considerable attention over the past decade [54, 44, 11, 55]. Our work is most related to [37], which uses an asynchronous nonlinear SA to study the finite-time convergence rate for asynchronous $Q$-learning on a single trajectory. Beyond [37], there are many other works that use SA schemes to study TD learning and $Q$-learning, e.g. [44, 52, 20]. The finite-time error bound for TD learning with state aggregation in our work is most related to the asymptotic convergence limit given in [49] and the application of SA scheme to asynchronous $Q$-learning in [37]. Beyond these papers, other related work in the broader area of RL with state aggregation includes [26, 23, 22, 9, 43]. We add to this literature with a novel finite-time convergence bound for a general SA with state aggregation. This result, in turn, yields the first finite-time error bound in the infinity norm for both TD learning with state aggregation and Q-learning with state aggregation.

## 2 Networked MARL

We consider a network of agents that are associated with an underlying undirected graph $\mathcal{G} = (\mathcal{N}, \mathcal{E})$, where $\mathcal{N} = \{1, 2, \cdots, n\}$ denotes the set of agents and $\mathcal{E} \subseteq \mathcal{N} \times \mathcal{N}$ denotes the set of edges. The distance $d_{\mathcal{G}}(i, j)$ between two agents $i$ and $j$ is defined as the number of edges on the shortest path that connects them on graph $\mathcal{G}$. Each agent is associated with its local state $s_i \in \mathcal{S}_i$ and local action $a_i \in \mathcal{A}_i$ where $\mathcal{S}_i$ and $\mathcal{A}_i$ are finite sets. The global state/action is defined as the combination of all local states/actions, i.e., $s = (s_1, \cdots, s_n) \in \mathcal{S} := \mathcal{S}_1 \times \cdots \times \mathcal{S}_n$, and $a = (a_1, \cdots, a_n) \in \mathcal{A} := \mathcal{A}_1 \times \cdots \times \mathcal{A}_n$. We use $N_i^{\kappa}$ to denote the $\kappa$-hop neighborhood of agent $i$ on $\mathcal{G}$, i.e., $N_i^{\kappa} := \{j \in \mathcal{N} \mid d_{\mathcal{G}}(i, j) \leq \kappa\}$. Let $f(\kappa) := \sup_i |N_i^{\kappa}|$. For a subset $M \subseteq \mathcal{N}$, we use $s_M/a_M$ to denote the tuple formed by the states/actions of agents in $M$.

Before we define the transitions and rewards, we first define the notion of active link sets, which are directed graphs on the agents $\mathcal{N}$ and they characterize the interaction structure among the agents. More specifically, an active link set is a set of directed edges that contains all self-loops, i.e., a subset of $\mathcal{N} \times \mathcal{N}$ and a super set of $\{(i, i) \mid i \in \mathcal{N}\}$. Generally speaking, $(j, i) \in L$ means agent $j$ can affect agent $i$ in the active link set $L$. Given an active link set $L$, we also use $N_i(L) := \{j \in \mathcal{N} \mid (j, i) \in L\}$ to denote the set of all agents (include itself) who can affect agent $i$ in the active link set $L$. In this paper, we consider a pair of active link sets $(L_t^s, L_t^r)$ that is independently drawn from some joint distribution $\mathcal{D}$ at each time step $t$,[1] where the distribution $\mathcal{D}$ will be defined using the underlying graph $\mathcal{G}$ later in Section 2.1. The role of $L_t^s/L_t^r$ is that they define the dependence structure of state transition/reward at time $t$, which we detail below.

*Transitions.* At time $t$, given the current state, action $s(t), a(t)$ and the active link set $L_t^s$, the next individual state $s_i(t + 1)$ is independently generated and only depends on the state/action of the agents in $N_i(L_t^s)$. In other words, we have,

$$P(s(t+1)|s(t), a(t), L_t^s) = \prod_{i \in \mathcal{N}} P_i(s_i(t+1)|s_{N_i(L_t^s)}(t), a_{N_i(L_t^s)}(t), L_t^s). \qquad (1)$$

*Rewards.* Each agent is associated with a local reward function $r_i$. At time $t$, it is a function of $L_t^r$ and the state/action of agents in $N_i(L_t^r)$: $r_i(L_t^r, s_{N_i(L_t^r)}(t), a_{N_i(L_t^r)}(t))$. The global reward $r(t)$ is defined to be the summation of the local rewards $r_i(t)$.

*Policy.* Each agent follows a localized policy that depends on its $\beta$-hop neighborhood, where $\beta \geq 0$ is a fixed integer. Specifically, at time step $t$, given the global state $s(t)$, agent $i$ adopts a local policy $\zeta_i$ parameterized by $\theta_i$ to decide the distribution of $a_i(t)$ based on the the states of agents in $N_i^\beta$.

Our objective is for all the agents to *cooperatively* maximize the discounted global reward, i.e., $J(\theta) = \mathbb{E}_{s \sim \pi_0}\left[\sum_{t=0}^{\infty} \gamma^t r(s(t), a(t)) \mid s(0) = s\right]$, where $\pi_0$ is a given distribution on the initial global state, and we recall $r(s(t), a(t))$ is the global stage reward defined as the sum of all local rewards at time $t$.

*Examples.* To highlight the applicability of the general model, we include two examples of networked systems that feature the dependence structure captured by our model in Appendix A: a wireless communication example and an example of controlling a process that spreads over a network.

Note that a limitation of our setting is that the dependence structure we consider is stationary, in the sense that dependencies are sampled i.i.d. from the distribution $\mathcal{D}$. It is important to consider more general time-varying forms (e.g. Markovian) in future research.

*Background.* Before moving on, we review a few key concepts in RL which will be useful in the rest of the section. We use $\pi_t^\theta$ to denote the distribution of $s(t)$ under policy $\theta$ given that $s(0) \sim \pi_0$. A well-known result [47] is that the gradient of the objective $\nabla J(\theta)$ can be computed by $\frac{1}{1-\gamma}\mathbb{E}_{s \sim \pi^\theta, a \sim \zeta^\theta(\cdot|s)}Q^\theta(s, a)\nabla \log \zeta^\theta(a \mid s)$, where distribution $\pi^\theta(s) = (1 - \gamma)\sum_{t=0}^{\infty}\gamma^t \pi_t^\theta(s)$ is the *discounted state visitation distribution*. Evaluating the $Q$-function $Q^\theta(s, a)$ plays a key role in approximating $\nabla J(\theta)$. The local $Q$-function for agent $i$ is the discounted local reward, i.e. $Q_i^\theta(s, a) = \mathbb{E}_{\zeta^\theta}\left[\sum_{t=0}^{\infty}\gamma^t r_i(t) \mid s(0) = s, a(0) = a\right]$, where we use $r_i(t)$ to denote the local reward of agent $i$ at time step $t$. Using local $Q$-functions, we can decompose the global $Q$-function as $Q^\theta(s, a) = \frac{1}{n}\sum_{i=1}^{n}Q_i^\theta(s, a)$, which allows each node to evaluate its local $Q$-function separately.

A key challenge in our MARL setting is that directly estimating the $Q$-functions is not scalable since the size of the $Q$-functions is exponentially large in the number of agents. Therefore, in Section 2.1, we study structural properties of the $Q$-functions resulting from the dependence structure in the transition (1), which enables us to design a scalable RL algorithm in Section 2.2.

## 2.1 $\mu$-decay Property

One of the core challenges for MARL is that the size of the $Q$ function is exponentially large in the number of agents. The key to our algorithm and its analysis is the identification of a novel structural

---

[1]Here, correlations between $L_t^s$ and $L_t^r$ are possible

decay property for the $Q$-function, which says that the local $Q$-function of each agent $i$ is mainly decided by the states of the agents who are near $i$. This property is critical for the design of scalable algorithms because it enables the agents to reduce the dimension of the $Q$-function by truncating its dependence of the states and actions of far away agents. Recently, exponential decay has been shown to hold in networked MARL when the network is static [38, 36], which is exploited to design a scalable RL algorithm. However, in stochastic network settings it is too much to hope for exponential decay in general [14], and so we introduce the more general notion of $\mu$-decay here, where $\mu$ is a function that converges to 0 as $\kappa$ tends to infinity. The case of exponential decay that has been studied previously corresponds to $\mu(\kappa) = \gamma^\kappa/(1-\gamma)$. The formal definition of $\mu$-decay is given below, where for simplicity, we use $i \xrightarrow{L} j$ to denote $(i,j) \in L$ and denote $N_{-i}^\kappa := \mathcal{N} \setminus N_i^\kappa$.

**Definition 2.1.** *For a function $\mu : \mathbb{N} \to \mathbb{R}^+$ that satisfies $\lim_{\kappa \to +\infty} \mu(\kappa) = 0$, the $\mu$-decay property holds if for any policy $\theta$ and any $i \in \mathcal{N}$, the local $Q$ function $Q_i^\theta$ satisfies $\left| Q_i^\theta(s,a) - Q_i^\theta(s',a') \right| \le \mu(\kappa)$ for any $(s,a),(s',a')$ that are identical within $N_i^\kappa$, i.e. $s_{N_i^\kappa} = s'_{N_i^\kappa}, a_{N_i^\kappa} = a'_{N_i^\kappa}$.*

Intuitively, if the $\mu$-decay property holds and $\mu(\kappa)$ decays quickly as $\kappa$ increases, we can approximately decompose the global $Q$ function as $Q^\theta(s,a) = \frac{1}{n}\sum_{i=1}^n Q_i^\theta(s,a) \approx \frac{1}{n}\sum_{i=1}^n \hat{Q}_i^\theta(s_{N_i^\kappa}, a_{N_i^\kappa})$, where $\hat{Q}_i$ only depends on the states and actions within the $\kappa$-hop neighborhood of agent $i$. Before our work, [46] empirically showed that such a value decomposition allows efficient training of MARL. Under the assumption that such decomposition exists, [46] propose an approach to learn this decomposition. In contrast, as we prove in this section, the $\mu$ decay property holds provably and therefore, the global $Q$ function can be directly decomposed in the networked MARL model and that the error of such decomposition is provably small.

Our first result is Theorem 2.1 which shows the relationship between the random active link sets and the $\mu$-decay property. The proof of Theorem 2.1 is deferred to Appendix B.1.

**Theorem 2.1.** *Define $L^a$ as the static active link set that contains all pairs $(i,j)$ whose graph distance on $\mathcal{G}$ is less than or equal to $\beta$, which is the dependency of local policy. Let random variable $X_i(\kappa)$ denote the smallest $t \in \mathbb{N}$ such that there exists a chain of agents*

$$j_0^a \xrightarrow{L_0^s} j_1^s \xrightarrow{L^a} j_1^a \xrightarrow{L_1^s} \dots \xrightarrow{L_{t-1}^s} j_t^s \xrightarrow{L^a} j_t^a,$$

*that satisfies $j_0^a \in N_{-i}^\kappa$ and $j_t^a \xrightarrow{L^r} i$. The $\mu$-decay property holds for $\mu(\kappa) = \frac{1}{1-\gamma}\mathbb{E}\left[\gamma^{X_i(\kappa)}\right]$.*

To make the $\mu$-decay result more concrete, we provide several scenarios that yield different upper bounds on the term $\mathbb{E}\left[\gamma^{X_i(\kappa)}\right]$. In the first scenario, we study the case where long range links do not exist in Corollary 2.2. In this case, we obtain an exponential decay property that generalizes the result in [38]. A proof is in Appendix B.2.

**Corollary 2.2** (Exponential Decay). *Consider a distribution $\mathcal{D}$ of active link sets that satisfies*

$$P_{(L^s,L^r)\sim\mathcal{D}}\{(i,j) \in L^s\} = 0, \text{ for all } i,j \in \mathcal{N} \text{ s.t. } d_\mathcal{G}(i,j) \ge \alpha_1,$$
$$P_{(L^s,L^r)\sim\mathcal{D}}\{(i,j) \in L^r\} = 0, \text{ for all } i,j \in \mathcal{N} \text{ s.t. } d_\mathcal{G}(i,j) \ge \alpha_2.$$

*Then, $\mathbb{E}\left[\gamma^{X_i(\kappa)}\right] \le C\rho^\kappa$, where $\rho = \gamma^{1/(\alpha_1+\beta)}, C = \gamma^{-\alpha_2/(\alpha_1+\beta)}$.*

In the second scenario, long range active links can occur, but with exponentially small probability with respect to their distance. In this case, we can obtain a near-exponential decay property where $\mu(\kappa) = O(\rho^{\kappa/\log\kappa})$ for some $\rho \in (0,1)$. A proof can be found in Appendix B.3.

**Theorem 2.3** (Near-Exponential Decay). *Suppose the distribution $\mathcal{D}$ of active link sets satisfies*

$$P_{(L^s,L^r)\sim\mathcal{D}}\{(i,j) \in L^s \cup L^r\} \le c\lambda^{d_\mathcal{G}(i,j)}, \text{ for all } i,j \in \mathcal{N},$$

*where $c \ge 1, 1 > \lambda > 0$ are constants. If the largest size of the $\kappa$ neighborhood in the underlying graph $\mathcal{G}$ can be bounded by a polynomial of $\kappa$, i.e., there exists some constants $c_0 \ge 1, n_0 \in \mathbb{N}$ such that $|\{j \in \mathcal{N} \mid d_\mathcal{G}(i,j) = \kappa\}| \le c_0(\kappa+1)^{n_0}$ holds for all $i$, then $\mathbb{E}\left[\gamma^{X_i(\kappa-1)}\right] \le C\rho^{\kappa/(1+\ln(\kappa+1))}$ for some positive constant $C$ and decay rate $\rho < 1$. [2]*

It is interesting to compare the result above with models of the so-called "small world phenomena" in social networks, e.g., [14]. In these models, a link $(i,j)$ occurs with probability $1/poly(d_\mathcal{G}(i,j))$,

---

[2]The explicit expression of $C$ and $\rho$ can be found in Appendix B.3.

---
**Algorithm 1** Scalable Actor Critic
---
1: **for** $m = 0, 1, 2, \cdots$ **do**
2:     Sample initial global state $s(0) \sim \pi_0$.
3:     Each node $i$ takes action $a_i(0) \sim \zeta_i^{\theta_i(m)}(\cdot \mid s_{N_i^\beta}(0))$ to obtain the global state $s(1)$.
4:     Each node $i$ records $s_{N_i^\kappa}(0), a_{N_i^\kappa}(0), r_i(0)$ and initialize $\hat{Q}_i^0$ to be all zero vector.
5:     **for** $t = 1, \cdots, T$ **do**
6:         Each node $i$ takes action $a_i(t) \sim \zeta_i^{\theta_i(m)}(\cdot \mid s_{N_i^\beta}(t))$ to obtain the global state $s(t+1)$.
7:         Each node $i$ update the local estimation $\hat{Q}_i$ with step size $\alpha_{t-1} = \frac{H}{t-1+t_0}$,

$$\hat{Q}_i^t\big(s_{N_i^\kappa}(t-1), a_{N_i^\kappa}(t-1)\big) =$$
$$(1-\alpha_{t-1})\hat{Q}_i^{t-1}\big(s_{N_i^\kappa}(t-1), a_{N_i^\kappa}(t-1)\big) + \alpha_{t-1}\Big(r_i(t) + \gamma\hat{Q}_i^{t-1}\big(s_{N_i^\kappa}(t), a_{N_i^\kappa}(t)\big)\Big),$$
$$\hat{Q}_i^t\big(s_{N_i^\kappa}, a_{N_i^\kappa}\big) = \hat{Q}_i^{t-1}\big(s_{N_i^\kappa}, a_{N_i^\kappa}\big) \text{ for } \big(s_{N_i^\kappa}, a_{N_i^\kappa}\big) \neq \big(s_{N_i^\kappa}(t-1), a_{N_i^\kappa}(t-1)\big).$$

8:     Each node $i$ approximate $\nabla_{\theta_i} J(\theta)$ by
        $\hat{g}_i(m) = \sum_{t=0}^T \gamma^t \frac{1}{n} \sum_{j \in N_i^\kappa} \hat{Q}_j^T\big(s_{N_j^\kappa}(t), a_{N_j^\kappa}(t)\big)\nabla_{\theta_i} \log \zeta_i^{\theta_i(m)}\big(a_i(t) \mid s_{N_i^\beta}(t)\big)$.
9:     Each node $i$ conducts gradient ascent by $\theta_i(m+1) = \theta_i(m) + \eta_m \hat{g}_i(m)$.
---

as opposed to the exponential dependence in Lemma 2.3. In this case, one can see function $\mu(\kappa)$ is lower bounded by $1/poly(\kappa)$, which leads us to conjecture that $\mu(\kappa)$ is also upper bounded by $O(1/poly(\kappa))$. Thus, when information spreads "slowly" it helps a localized algorithm to learn efficiently.

## 2.2 A Scalable Actor Critic Algorithm

Motivated by the $\mu$-decay property of the $Q$-functions, we design a novel Scalable Actor Critic algorithm (Algorithm 1) for networked MARL problem, which exploits the $\mu$-decay result in the previous section. The Critic part (from line 2 to line 7) uses the local trajectory $\{(s_{N_i^\kappa}, a_{N_i^\kappa}, r_i)\}$ to evaluate the local $Q$-functions under parameter $\theta(m)$. Intuitively, the $\mu$-decay property guarantees that we can achieve good approximation error even when $\kappa$ is not large. The Actor part (from line 8 to line 9) computes the estimated partial derivative using the estimated local $Q$-functions, and uses the partial derivative to update local parameter $\theta_i$. The step size sequence $\{\eta_m\}$ will be defined in Theorem B.2. Compared with the Scalable Actor Critic algorithm proposed in [38], Algorithm 1 extends the policy dependency structure considered. No longer is the dependency completely local; it now extends to all agents within the $\beta$-hop neighborhood. Interestingly, the time-varying dependencies do not add complexity into the algorithm (though the analysis is more complex).

Algorithm 1 is highly scalable. Each agent $i$ needs only to query and store the information within its $\kappa$-hop neighborhood during the learning process. The parameter $\kappa$ can be set to balance accuracy and complexity. Specifically, as $\kappa$ increases, the error bound becomes tighter at the expense of increasing computation, communication, and space complexity.

## 2.3 Convergence

We now present our main result, a finite-time error bound for the Scalable Actor Critic algorithm (Algorithm 1) that holds under general (non-local) dependencies. To that end, we first describe the assumption needed in our result. It focuses on the Markov chain formed by the global state-action pair $(s, a)$ under a fixed policy parameter $\theta$ and is standard for finite-time convergence results in RL, e.g., [44, 5, 37].

**Assumption 2.1.** *Under any fixed policy $\theta$, $\{z(t) := (s(t), a(t))\}$ is an aperiodic and irreducible Markov chain on state space $\mathcal{Z} := \mathcal{S} \times \mathcal{A}$ with a unique stationary distribution $d^\theta = (d_z^\theta, z \in \mathcal{Z})$, which satisfies $d_z^\theta > 0, \forall z \in \mathcal{Z}$. Define $d^\theta(z') = \sum_{z \in \mathcal{Z}: z_{N_i^\kappa} = z'} d^\theta(z)$ and $\sigma'(\kappa) := \inf_{z' \in \mathcal{Z}_{N_i^\kappa}} d^\theta(z')$. There exists positive constants $K_1, K_2$ such that $K_2 \geq 1$ and $\forall z' \in \mathcal{Z}, \forall t \geq 0, \sup_{\mathcal{K} \subseteq \mathcal{Z}} \left| \sum_{z \in \mathcal{K}} d_z^\theta - \sum_{z \in \mathcal{K}} \mathbb{P}(z(t) = z \mid z(0) = z') \right| \leq K_1 e^{-t/K_2}.$*

We next analyze the Critic part of Algorithm 1 within a given outer loop iteration $m$. Since the policy is fixed in the inner loop, the global state/action pair $(s, a)$ in the original MDP can be viewed as the state of a Markov chain. We observe that each local estimate $\hat{Q}_i^t(s_{N_i^\kappa}, a_{N_i^\kappa})$ can be viewed as a form of state aggregation, where the global state $(s, a)$ is "compressed" to $h(s, a) := (s_{N_i^\kappa}, a_{N_i^\kappa})$. Broadly speaking, the technique of state aggregation is one of the easiest-to-deploy schemes for state space compression [21, 43], while its final performance relies heavily on whether the state aggregation map $h$ only aggregates "similar" states. To have a good approximate equivalence, we need to find a good $h$, i.e., if two states are mapped to the same abstract state, their value functions are required to be close (to be discussed in Theorem 3.2). In the context of networked MARL, the $\mu$ decay property (Definition 2.1) provides a natural mapping for state aggregation $h(s, a) := (s_{N_i^\kappa}, a_{N_i^\kappa})$ which we defined earlier. This mapping $h$ maps the global state/action to the local states/actions in agent $i$'s $\kappa$-hop neighborhood and the $\mu$-decay property guarantees that if $h(s, a) = h(s', a')$, the difference in their $Q$-functions is upper bounded by $\mu(\kappa)$, which is vanishing as $\kappa$ increases. This shows that the mapping $h$ we used is "good" in the sense it aggregates very similar global state-action pairs. This idea leads to the following theorem about the Critic part of Scalable Actor Critic (Algorithm 1).

**Theorem 2.4.** *Suppose Assumption 2.1 and $\mu$-decay property (Definition 2.1) hold. Let the step size be $\alpha_t = \frac{H}{t+t_0}$ with $t_0 = \max(4H, 2K_2 \log T)$, and $H \geq \frac{2}{(1-\gamma)\sigma'(\kappa)}$. Define constant $C_b := 4K_1(1 + 2K_2 + 4H)$. Then, inside outer loop iteration $m$, for each $i \in \mathcal{N}$, with probability at least $1 - \delta$, we have $\sup_{(s,a) \in \mathcal{S} \times \mathcal{A}} \left| Q_i^{\theta(m)}(s, a) - \hat{Q}_i^T(s_{N_i^\kappa}, a_{N_i^\kappa}) \right| \leq \frac{C_a}{\sqrt{T+t_0}} + \frac{C_a'}{T+t_0} + \frac{\mu(\kappa)}{1-\gamma}$, where the constants are given by $C_a = \frac{40H}{(1-\gamma)^2} \sqrt{K_2 \log T \left( \log\left( \frac{4f(\kappa)K_2 T}{\delta} \right) + \log \log T \right)}$ and $C_a' = \frac{8}{(1-\gamma)^2} \max\left\{ \frac{144K_2 H \log T}{\sigma'(\kappa)} + C_b, 2K_2 \log T + t_0 \right\}$.*

The proof of Theorem 2.4 can be found in Appendix B.4. The most related result in the literature to Theorem 2.4 is Theorem 7 in [38]. In comparison, Theorem 2.4 applies for more general, potentially non-local, dependencies and, also, improves the constant term by a factor of $1/(1 - \gamma)$.

To analyze the Actor part of Algorithm 1, we make the following additional boundedness and Lipschitz continuity assumptions on the gradients. These are standard assumptions in the literature.

**Assumption 2.2.** *For any $i, a_i, s_{N_i^\beta}$ and $\theta_i$, we assume $\left\| \nabla_{\theta_i} \log \zeta_i^{\theta_i}(a_i \mid s_{N_i^\beta}) \right\| \leq W_i$. Then, for any $L_t^a$, $\left\| \nabla_\theta \log \zeta^\theta(a \mid s) \right\| \leq W := \sqrt{\sum_{i=1}^n W_i^2}$. We further assume $\nabla J(\theta)$ is $W'$-Lipschitz in $\theta$.*

Intuitively, since the quality of the estimated policy gradient depends on the quality of the estimation of $Q$-functions, if every agent $i$ has learned a good approximation of its local $Q$-function in the Critic part of Algorithm 1, the policy gradient can be approximated well. Therefore, the Actor part can obtain a good approximation of a stationary point of the objective function. We state the sample complexity result in Theorem 2.5 and defer the detailed bounds and a proof to Appendix B.5.

**Theorem 2.5.** *Under Assumption 2.2, to reach an $O(\epsilon)$-approximate stationary point with probability at least $1 - \delta$, we need to choose $\kappa$ such that $\mu(\kappa) = O(W^{-2}(1-\gamma)^4 \epsilon)$. The number of required iterations of the outer loop should satisfy $M = \tilde{\Omega}(\epsilon^{-2} poly(W, W', \frac{1}{1-\gamma}))$ and the number of required iterations of the inner loop is $T = \tilde{\Omega}(\epsilon^{-2} poly(W, \frac{1}{\sigma'(\kappa)}, K_2, \frac{1}{1-\gamma}, \log f(\kappa), \log(1/\delta)))$.*

Note that $W$ scales with the number of agents $n$. Thus, Theorem 2.5 shows that the complexity of our algorithm scales with the largest state-action space size of any $\kappa$-hop neighborhood and the number of agents $n$, which avoids the exponential blowup in $n$ when the graph is sparse and achieves scalable RL for networked agents even under stochastic, non-local settings.

## 3 Proof Idea: Stochastic Approximation and State Aggregation

In this section, we present the key technical innovation underlying our results on MARL in Theorem 2.4: a new finite-time analysis of a general asynchronous stochastic approximation (SA) scheme. As we mention in Section 2, the truncation enabled by $\mu$-decay provides a form of state aggregation, which we analyze via a general SA scheme in Section 3.1. Further, this SA scheme is of interest more broadly, e.g., to the settings of TD learning with state aggregation (Section 3.2) and asynchronous $Q$-learning with state aggregation (Appendix D.4).

## 3.1 Stochastic Approximation

Consider a finite-state Markov chain whose state space is given by $\mathcal{N} = \{1, 2, \cdots, n\}$. Let $\{i_t\}_{t=0}^{\infty}$ be the sequence of states visited by this Markov chain. Our focus is generalizing the following asynchronous stochastic approximation (SA) scheme, which is studied in [48, 41, 52]: Let parameter $x \in \mathbb{R}^{\mathcal{N}}$, and $F : \mathbb{R}^{\mathcal{N}} \to \mathbb{R}^{\mathcal{N}}$ be a $\gamma$-contraction in the infinity norm. The update rule of the SA scheme is given by

$$
\begin{aligned}
x_{i_t}(t+1) &= x_{i_t}(t) + \alpha_t(F_{i_t}(x(t)) - x_{i_t}(t) + w(t)), \\
x_j(t+1) &= x_j(t) \text{ for } j \neq i_t, j \in \mathcal{N},
\end{aligned} \tag{2}
$$

where $w(t)$ is a noise sequence. It is shown in [37] that parameter $x(t)$ converges to the unique fixed point of $F$ at the rate of $O(1/\sqrt{t})$.

While general, in many cases, including networked MARL, we do not wish to calculate an entry for every state in $\mathcal{N}$ in parameter $x$, but instead, wish to calculate "aggregated entries." Specifically, at each time step, after $i_t$ is generated, we use a surjection $h$ to decide which dimension of parameter $x$ should be updated. This technique, referred to as state aggregation, is one of the easiest-to-deploy schemes for state space compression in the RL literature [21, 43]. In the generalized SA scheme, our objective is to specify the convergence point as well as obtain a finite-time error bound.

Formally, to define the generalization of (2), let $\mathcal{N} = \{1, \cdots, n\}$ be the state space of $\{i_t\}$ and $\mathcal{M} = \{1, \cdots, m\}, (m \leq n)$ be the *abstract* state space. The surjection $h : \mathcal{N} \to \mathcal{M}$ is used to convert every state in $\mathcal{N}$ to its abstraction in $\mathcal{M}$. Given parameter $x \in \mathbb{R}^{\mathcal{M}}$ and function $F : \mathbb{R}^{\mathcal{N}} \to \mathbb{R}^{\mathcal{N}}$, we consider the generalized SA scheme that updates $x(t) \in \mathbb{R}^{\mathcal{M}}$ starting from $x(0) = \mathbf{0}$,

$$
\begin{aligned}
x_{h(i_t)}(t+1) &= x_{h(i_t)}(t) + \alpha_t\big(F_{i_t}(\Phi x(t)) - x_{h(i_t)}(t) + w(t)\big), \\
x_j(t+1) &= x_j(t) \text{ for } j \neq h(i_t), j \in \mathcal{M},
\end{aligned} \tag{3}
$$

where the feature matrix $\Phi \in \mathbb{R}^{\mathcal{N} \times \mathcal{M}}$ is defined as

$$
\Phi_{ij} = \begin{cases} 1 & \text{if } h(i) = j \\ 0 & \text{otherwise} \end{cases}, \forall i \in \mathcal{N}, j \in \mathcal{M}. \tag{4}
$$

In order to state our main result characterizing the convergence of (3), we must first state a few definitions and assumptions. To begin, we define the weighted infinity norm as in [37], except that we extend its definition so as to define the contraction of function $F$. The reason we use the weighted infinity norm as opposed to the standard infinity norm is that its generality can be used in certain settings for undiscounted RL, as shown in [48, 2].

**Definition 3.1** (Weighted Infinity Norm). *Fix a positive vector $v \in \mathbb{R}^{\mathcal{M}}$. For $x \in \mathbb{R}^{\mathcal{M}}$, we define* $\|x\|_v := \sup_{i \in \mathcal{M}} \frac{|x_i|}{v_i}$. *For $x \in \mathbb{R}^{\mathcal{N}}$, we define* $\|x\|_v := \sup_{i \in \mathcal{N}} \frac{|x_i|}{v_{h(i)}}$.

Next, we state our assumption on the mixing rate of the Markov chain $\{i_t\}$, which is common in the literature [50, 44]. It holds for any finite-state Markov chain which is aperiodic and irreducible [5].

**Assumption 3.1** (Stationary Distribution and Geometric Mixing Rate). *$\{i_t\}$ is an aperiodic and irreducible Markov chain on state space $\mathcal{N}$ with stationary distribution $d = (d_1, d_2, \cdots, d_n)$. Let $d'_j = \sum_{i \in h^{-1}(j)} d_i$ and $\sigma' = \inf_{j \in \mathcal{M}} d'_j$. There exists positive constants $K_1, K_2$ which satisfy that $\sup_{\mathcal{S} \subseteq \mathcal{N}} \left| \sum_{i \in \mathcal{S}} d_i - \sum_{i \in \mathcal{S}} \mathbb{P}(i_t = i \mid i_0 = j) \right| \leq K_1 \exp(-t/K_2), \forall j \in \mathcal{N}, \forall t \geq 0$ and $K_2 \geq 1$.*

Our next assumption ensures contraction of $F$. It is also standard, e.g., [48, 52, 37], and ensures that $F$ has a unique fixed point $y^*$.

**Assumption 3.2** (Contraction). *Operator $F$ is a $\gamma$ contraction in $\|\cdot\|_v$, i.e., for any $x, y \in \mathbb{R}^{\mathcal{N}}$, we have $\|F(x) - F(y)\|_v \leq \gamma \|x - y\|_v$. Further, there exists some constant $C > 0$ such that for any $x \in \mathbb{R}^{\mathcal{N}}$, we have $\|F(x)\|_v \leq \gamma \|x\|_v + C$.*

In Assumption 3.2, notice that the first sentence directly implies the second with $C = (1+\gamma)\|y^*\|_v$, where $y^* \in \mathbb{R}^{\mathcal{N}}$ is the unique fixed point of $F$. Further, while Assumption 3.2 implies that $F$ has a unique fixed point $y^*$, we do not expect our stochastic approximation scheme to converge to it. Instead, we show that the convergence is to the unique $x^*$ that solves

$$
\Pi F(\Phi x^*) = x^*, \text{ where } \Pi := \left(\Phi^\top D \Phi\right)^{-1} \Phi^\top D. \tag{5}
$$

Here $D = diag(d_1, d_2, \cdots, d_n)$ denotes the steady-state probabilities for the process $\{i_t\}$. Note that $x^*$ is well-defined because the operator $\Pi F(\Phi \cdot)$, which defines a mapping from $\mathbb{R}^{\mathcal{M}}$ to $\mathbb{R}^{\mathcal{M}}$, is also a contraction in $\|\cdot\|_v$. We state and prove this as Proposition C.1 in Appendix C.1.

Our last assumption is on the noise sequence $w(t)$. It is also standard, e.g., [41, 37].

**Assumption 3.3** (Martingale Difference Sequence). *$w_t$ is $\mathcal{F}_{t+1}$ measurable and satisfies $\mathbb{E}w(t) \mid \mathcal{F}_t = 0$. Further, $|w(t)| \leq \bar{w}$ almost surely for constant $\bar{w}$.*

We are now ready to state our finite-time convergence result for stochastic approximation.

**Theorem 3.1.** *Suppose Assumptions 3.1, 3.2, 3.3 hold. Further, assume there exists constant $\bar{x} \geq \|x^*\|_v$ such that $\forall t, \|x(t)\|_v \leq \bar{x}$ almost surely.[3] Let the step size be $\alpha_t = \frac{H}{t + t_0}$ with $t_0 = \max(4H, 2K_2 \log T)$, and $H \geq \frac{2}{\sigma'(1-\gamma)}$. Let $x^*$ be the unique solution of equation $\Pi F(\Phi x^*) = x^*$, and define constants $C_1 := 2\bar{x} + C + \frac{\bar{w}}{\underline{v}}, C_2 := 4\bar{x} + 2C + \frac{\bar{w}}{\underline{v}}, C_3 := 2K_1(2\bar{x} + C)(1 + 2K_2 + 4H)$. Then, with probability at least $1 - \delta$,*

$$\|x(T) - x^*\|_v \leq \frac{C_a}{\sqrt{T + t_0}} + \frac{C_a'}{T + t_0} = \tilde{O}\left(\frac{1}{\sqrt{T}}\right),$$

*where the constants are given by $C_a = \frac{4HC_2}{1-\gamma}\sqrt{K_2 \log T\left(\log\left(\frac{4mK_2T}{\delta}\right) + \log\log T\right)}$ and $C_a' = 4\max\left\{\frac{48K_2C_1 H \log T + \sigma' C_3}{(1-\gamma)\sigma'}, \frac{2\bar{x}(2K_2 \log T + t_0)}{1-\gamma}\right\}$.*

A proof of Theorem 3.1 can be found in Appendix C.2. Compared with Theorem 4 in [37], Theorem 3.1 holds for a more general SA scheme where state aggregation is used to reduce the dimension of the parameter $x$. The proof technique used in [37] does not apply to our setting because our stationary point $x^*$ has a more complex form (4). To do the generalization, we need to use a different error decomposition method compared to [37] that leverages the stationary distribution $D$ rather than the distribution of $i_t$ condition on $i_{t-\tau}$ (see Appendix C.2 for details). Because of this generality, Theorem 3.1 requires a stronger but standard assumption on the mixing rate of the Markov chain $\{i_t\}$.

## 3.2 State Aggregation

To illustrate the impact of our analysis of SA (Theorem 3.1) beyond the network setting, we study a simpler application to the cases of TD-learning and $Q$-learning with state aggregation in this section. Understanding state aggregation methods is a foundational goal of analysis in the RL literature and it has been studied in many previous works, e.g., [26, 23, 22, 9, 43]. Further, the result is extremely useful in the analysis in networked MARL that follows since the $\mu$-decay property we introduce (Definition 2.1) provides a natural state aggregation in the network setting (see Corollary 2.4). Due to space constraints, in this section we only introduce the results on TD-learning; the results on $Q$-learning are given in Appendix D.4.

In TD learning with state aggregation [43, 49], given the sequence of states visited by the Markov chain is $\{i_t\}$, the update rule of TD(0) is given by

$$\theta_{h(i_t)}(t + 1) = \theta_{h(i_t)}(t) + \alpha_t\left(r_t + \gamma\theta_{h(i_{t+1})}(t) - \theta_{h(i_t)}(t)\right),$$
$$\theta_j(t + 1) = \theta_j(t) \text{ for } j \neq h(i_t), j \in \mathcal{M},$$
(6)

where $h : \mathcal{N} \to \mathcal{M}$ is a surjection that maps each state in $\mathcal{N}$ to an abstract state in $\mathcal{M}$ and $r_t$ is the reward at time step $t$ such that $\mathbb{E}[r_t] = r(i_t, i_{t+1})$.

Taking $F$ as the Bellman Policy Operator, i.e., the $i$'th dimension of function $F$ is given by

$$F_i(V) = \mathbb{E}_{i' \sim \mathbb{P}(\cdot|i)}[r(i, i') + \gamma V_{i'}], \forall i \in \mathcal{N}, V \in \mathbb{R}^{\mathcal{N}}.$$

The value function (vector) $V^*$ is defined as $V_i^* = \mathbb{E}[\sum_{t=0}^{\infty} \gamma^t r(i_t, i_{t+1}) \mid i_0 = i], i \in \mathcal{N}$ [49]. By defining the feature matrix $\Phi$ as (4) and the noise sequence as

$$w(t) = r_t + \gamma\theta_{h(i_{t+1})}(t) - \mathbb{E}_{i' \sim \mathbb{P}(\cdot|i_t)}[r(i_t, i') + \gamma\theta_{h(i')}(t)],$$

we can rewrite the update rule of TD(0) in (6) in the form of an SA scheme (3). Therefore, we can apply Theorem 3.1 to obtain a finite-time error bound for TD learning with state aggregation. A proof of Theorem 3.2 can be found in Appendix D.2.

---

[3]The assumption on $\bar{x}$ follows from Assumptions 3.2 and 3.3. See Proposition C.2 in Appendix C.3.

**Theorem 3.2.** *Let Assumption 3.1 hold for the Markov chain $\{i_t\}$ and let the stage reward $r_t$ be upper bounded by $\bar{r}$ almost surely. Assume that if $h(i) = h(i')$ for $i, i' \in \mathcal{N}$, we have $|V_i^* - V_{i'}^*| \le \zeta$ for a constant $\zeta$. Consider TD(0) with the step size $\alpha_t = \frac{H}{t+t_0}$, where $t_0 = \max(4H, 2K_2 \log T)$ and $H \ge \frac{2}{\sigma'(1-\gamma)}$. Define constant $C_4 := 4K_1(1 + 2K_2 + 4H)$. Then, with probability at least $1 - \delta$,*

$$\|\Phi \cdot \theta(T) - V^*\|_\infty \le \frac{C_a}{\sqrt{T + t_0}} + \frac{C_a'}{T + t_0} + \frac{\zeta}{1 - \gamma},$$

*where the constants are given by $C_a = \frac{40 H \bar{r}}{(1-\gamma)^2} \sqrt{K_2 \log T \left(\log\left(\frac{4mK_2 T}{\delta}\right) + \log \log T\right)}$ and $C_a' = \frac{8\bar{r}}{(1-\gamma)^2} \max\{\frac{144 K_2 H \log T}{\sigma'} + C_4, 2K_2 \log T + t_0\}$.*

The most related prior results to Theorem 3.2 are [44, 4]. In contrast to these, Theorem 3.2 considers the infinity norm, which is more natural for measuring error when using state aggregation. Further, our analysis is different and extends to the case of $Q$-learning with state aggregation (see Appendix D.4), where we obtain the first finite-time error bound. Moreover, unlike [4], our TD-learning algorithm does not require a projection step.

## 4   Concluding Remarks

In this paper, we propose and analyze the Scalable Actor Critic Algorithm that provably learns a near-optimal local policy in a setting where every agent is allowed to interact with a random subset of agents. The $\mu$-decay property, which enables the decentralized approximation of local $Q$ functions, is the key to our approach.

There are a number of future directions motivated by the results in this paper. For example, we allow the interaction structure among the agents to change in a stochastic way in this work. It is interesting to see if such structure can be time-varying in more general ways (e.g., Markovian or adversarial). Besides, although our Scalable Actor Critic algorithm consumes much less memory than a centralized tabular approach, the memory space required by each agent $i$ to store $\hat{Q}_i$ grows exponentially with respect to $f(\kappa)$, which denotes the size of the largest $\kappa$-hop neighborhood. Thus, memory problems may still arise if $f$ grows quickly as $\kappa$ increases. Therefore, an interesting open problem is whether we can apply additional function approximations on truncated state/action pair $\left(s_{N_i^\kappa}, a_{N_i^\kappa}\right)$, and obtain similar finite-time convergence guarantees as Scalable Actor Critic.

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
