# A  Examples

## A.1  Wireless Networks

We consider a wireless network with multiple access points setting shown in Fig. 1, where a set of user nodes in a wireless network, denoted by $U = \{u_1, u_2, \cdots, u_n\}$, share a set of access points $Y = \{y_1, y_2, \cdots, y_m\}$ [60]. Each access point $y_i$ is associated with a probability $p_i$ of successful transmission. Each user node $u_i$ only has access to a subset $Y_i \subseteq Y$ of the access points. Typically, this available set is determined by each user node's physical connections to the access points. To apply the networked MARL model, we identify the set of user nodes $U$ as the set of agents $\mathcal{N}$ in Section 2. The underlying graph $G = (\mathcal{N}, \mathcal{E})$ is defined as the conflict graph, i.e., edge $(u_i, u_j) \in \mathcal{E}$ if and only if $Y_i \cap Y_j \neq \emptyset$.

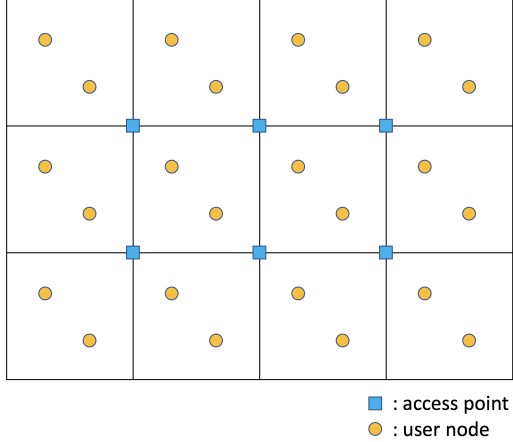

■ : access point
● : user node

Figure 1: An example setup of wireless networks. Each user node can send packets to the access points at the corners of its grid.

At each time step $t$, each user $u_i$ receives a packet with initial life span $d$ with probability $q$. Each user maintains a queue to cache the packets it receives. At each time step, if the packet is successfully sent to an access point, it will be removed from the queue. Otherwise, its life span will decrease by 1. A packet is discarded from the queue immediately if its remaining life span is 0. At each time step $t$, a user node $u_i$ can choose to send one of the packets in its queue to one of the access point $y_{i,t} \in Y_i$. If no other user node sends packets to access point $y_{i,t}$ at time step $t$, the packet from user $i$ can be delivered successfully with probability $p_i$. Otherwise, the sending action will fail. A user $u_i$ receives a local reward of $r_{i,t} = 1$ immediately after successfully sending a packet at time step $t$, and receives $r_{i,t} = 0$ otherwise. Our objective is to find a policy that maximizes the global discounted reward under a discounted factor $0 \leq \gamma < 1$:

$$\mathbb{E}\left[\sum_{i=1}^{n}\sum_{t=0}^{\infty}\gamma^t r_{i,t}\right].$$

To see how this setting fits into our model, we first define the local state/action and specify the parameters. Since each packet has a life span of $d$, and each user node receives at most one packet at a time step, we use a $d$-tuple $s_i = (e_1, e_2, \cdots, e_d) \in \mathcal{S}_i := \{0,1\}^d$ to denote the local state of user node $i$. Specifically, $e_j$ indicates whether user node $u_i$ has a packet with remaining life span $j$ in its queue. A local action of user node $u_i$ is 2-tuple $(l, y)$, which means sending the packet with remaining life span $l \in \{1, 2, \cdots, d\}$ to an access point $y \in Y_i$. Note that we define an empty action that does nothing at all. If a user node performs an action $(l, y)$ when there is no packet with life span $l$ in its queue, we view this as an empty action. This setting falls into the category we studied in Corollary 2.2, where long range links do not exist. Specifically, in this setting, the next local state of user node $u_i$ depends on the current local states/actions in its 1-hop neighborhood ($\alpha_1 = 1$ in Corollary 2.2). We assume each user node can choose its action only based on its current local state ($\beta = 0$). Due to potential collisions, the local reward of user $u_i$ also depends on the states/actions in

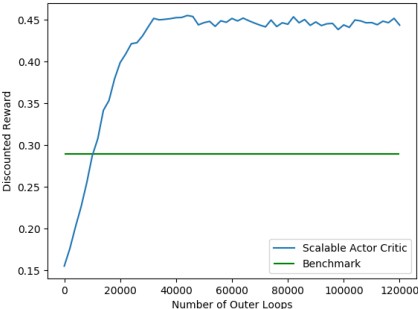

Figure 2: Discounted reward in the training process. $5 \times 5$ grid, 1 user per grid.

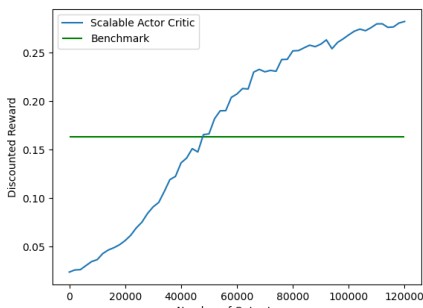

Figure 3: Discounted reward in the training process. $3 \times 4$ grid, 2 users per grid.

its 1-hop neighborhood ($\alpha_2 = 1$ in Corollary 2.2). Though this is a static setting, note that the results of [38] do not apply.

The detailed setting we use is as follows. We consider the setting where the user nodes are located in $h \times w$ grids (see Fig. 1). There are $c$ user nodes in each grid, and each user can send packets to an access point on the corner of its grid. We set the initial life span $d = 2$, the arrival probability $q = 0.5$, and the discounted factor $\gamma = 0.7$. The successful transmission probability $p_i$ for each access point $y_i$ is sampled uniformly randomly from $[0, 1]$. We run the Scalable Actor Critic algorithm with parameter $\kappa = 1$ to learn a localized stochastic policy in two cases $(h, w, c) = (5, 5, 1)$ (see Fig. 2) and $(h, w, c) = (3, 4, 2)$ (see Fig. 3). For comparison, we use a benchmark based on the localized ALOHA protocol [39]. Specifically, the benchmark policy works as following: At time step $t$, each user node $u_i$ takes the empty action with a certain probability $p'$; otherwise, it sends the packet with the minimum remaining life span to a random access point in $Y_i$, with the probability proportional to the successful transmission probability of this access point and inverse proportional to the number of users sharing this access point. In Fig. 2 and Fig. 3, we have tuned the parameter $p'$ to find the one with the highest discounted reward.

As shown in Fig. 2 and Fig. 3, starting from the initial policy that chooses an local action uniformly at random, the Scalable Actor Critic algorithm with parameter $\kappa = 1$ can learn a policy that performs better than the benchmark. As a remark, the benchmark policy requires the set $\{p_i\}_{1 \leq i \leq m}$, the probability of successful transmission, as input. Moreover, in the benchmark policy, the probability of performing an empty action also needs to be tuned manually. In contrast, the Scalable Actor Critic algorithm can learn a better policy without these specific inputs by interacting with the system.

## A.2 Spreading Networks

We consider a spreading network with $n$ agents and an underlying graph $\mathcal{G}$. See Fig. 4 for an illustration of $n = wh$ agents on a $w \times h$ grid network. For each agent $i$, the local state/action space is given by $\mathcal{S}_i = \{0, 1\}$ and $\mathcal{A}_i = \{0, 1\}$. To make the discussion more concrete, in the following we present the spreading network model in the context of SIS epidemic network. This version of the SIS model has been studied in, for example, [40]. Our setting is more general and can be generalized to other types of spreading networks like opinion networks, social networks, etc. At time step $t$, the local state $s_i(t) = 0$ means agent $i$ is "susceptible", while the local state $s_i(t) = 1$ means the agent $i$ is "infected". By taking action $a_i(t) = 1$, agent $i$ can suppress its infection probability at the expense of incurring an action cost. In the meantime, agent $i$ will incur an infection cost if $s_i(t) = 1$. The interaction among agents is modeled by a set of undirected links, where two agents can affect each other if they are connected by a link. To model the influence of physical distance on the pattern of social contact, we assume the short range links occur more frequently than long range links. An illustration of the spreading network is shown in Fig. 4 (a), where the black nodes denote the agents with state 1; the white nodes denote the agents with state 0; the blue edges denote the set of active links at some time step.

Mathematically, the model can be described as follows. At each time step $t$, each agent $i$ can decide her/his local action $a_i(t)$ based on the information of local states in the 1-hop neighborhood $N_i^1$, i.e.,

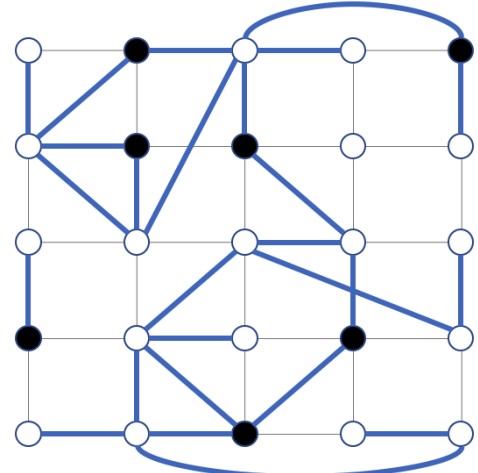

Figure 4: An illustration of the spreading network with 25 agents on a $5 \times 5$ grid network. The black nodes denote "infected" agents; The white nodes denote "susceptible" agents; The blue edges denote the active links at some time step.

$\beta = 1$. The local reward $r_i(t)$ is a function of the local state $s_i(t)$ and the local action $a_i(t)$, i.e., $L_t^r$ is static and only contains self loops. Specifically, we define

$$r_i(t) = -c_i^{(a)} \mathbf{1}(a_i(t) = 1) - c_i^{(s)} \mathbf{1}(s_i(t) = 1),$$

where $\left(c_i^{(s)}, c_i^{(a)}\right)$ are parameters associated with agent $i$ and can be different among agents. As mentioned earlier, $c_i^{(s)}$ penalizes the agent for being "infected", while $c_i^{(a)}$ is the cost of taking epidemic control measure. The stage reward is the sum of these two costs.

To describe the state transition rule, we first define the way the active link set $L_t^s$ is generated: independently for each pair of agents $(i, j) \in \mathcal{N} \times \mathcal{N}$ with $i \neq j$, with probability $2^{-d_\mathcal{G}(i,j)}$, we include edges $(i, j)$ and $(j, i)$ in the set $L_t^s$; otherwise, neither edge is included in the set, i.e. $(i, j), (j, i) \notin L_t^s$. Given $L_t^s$, the next local state $s_i(t+1)$ is sampled from a distribution that depends on the local states in $N_i(L_t^s)$. Specifically, define the quantities

$$n_i(t) = |\{j \mid j \in N_i(L_t) \setminus \{i\}, s_j(t) = 1, a_j(t) = 0\}|,$$
$$m_i(t) = |\{j \mid j \in N_i(L_t) \setminus \{i\}, s_j(t) = 1, a_j(t) = 1\}|.$$

Then, the probability that $s_i(t+1) = 0$ is given by

$$P(s_i(t+1) = 0 \mid s_{N_i(L_t)}, a_{N_i(L_t)}) = \begin{cases} p_i^{(r)} & \text{if } s_i(t) = 1; \\ \left(1 - p_i^{(h)}\right)^{n_i(t)} \left(1 - p_i^{(m)}\right)^{m_i(t)} & \text{if } s_i(t) = 0, a_i(t) = 1; \\ \left(1 - p_i^{(m)}\right)^{n_i(t)} \left(1 - p_i^{(l)}\right)^{m_i(t)} & \text{if } s_i(t) = 0, a_i(t) = 0, \end{cases}$$

where $\left(p_i^{(r)}, p_i^{(h)}, p_i^{(m)}, p_i^{(l)}\right)$ are parameters associated with agent $i$ and can be different among agents. Due to control actions, we assume $p_i^{(h)} > p_i^{(m)} > p_i^{(l)}$. This provides the transition rule, and the underlying intuition is that the local state of agent $i$ turns from "infected" ($s_i(t) = 1$) to "susceptible" ($s_i(t+1) = 0$) with a fixed recovering probability $p_i^{(r)}$; the probability that agent $i$ turns from "susceptible" ($s_i(t) = 0$) to "infected" ($s_i(t+1) = 1$) depends on the number of neighboring agents in the active link set that are already infected, and further, whether agent $i$ or the nearby agents $j$ take epidemic control measures ($a_i(t) = 1, a_j(t) = 1$) or not. Roughly speaking, the more nearby infected agents, the more likely agent $i$ will become infected; however, if epidemic control measures are taken by agent $i$ and nearby agents in $N_i(L_t^s)$, the probability of agent $i$ getting infected will be smaller.

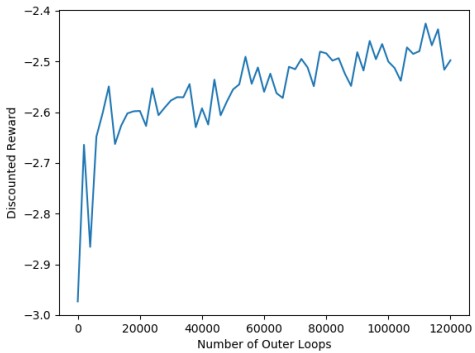

Figure 5: Discounted reward in the training process. $5 \times 5$ grid.

We run the Scalable Actor Critic algorithm with parameter $\kappa = 1$ to learn a localized stochastic policy in the case $(h, w) = (5, 5)$ (Fig. 5). For each agent $i$, parameters $\left( c_i^{(s)}, c_i^{(a)}, p_i^{(r)}, p_i^{(h)} \right)$ are sampled independently from the distribution

$$c_i^{(s)} \sim U[1.0, 3.0], c_i^{(a)} \sim U[0.01, 0.20], p_i^{(r)} \sim U[0.1, 0.5], p_i^{(h)} \sim U[0.5, 0.9],$$

and we set $p_i^{(m)} = p_i^{(h)}/4, p_i^{(l)} = p_i^{(m)}/4$. At time step 0, for each $i \in \mathcal{N}$, we initialize local state $s_i(0)$ to be 1 with probability 0.3.

## B   Stochastic Networked MARL

### B.1   Proof of Theorem 2.1

For ease of exposition, let $A, B$ be two subsets of the agent set $\mathcal{N}$ and we use $A \xrightarrow{\tau} B$ to denote the event that there exists a chain

$$j_0^a \xrightarrow{L_0^s} j_1^s \xrightarrow{L^a} j_1^a \xrightarrow{L_1^s} \dots \xrightarrow{L_{\tau-1}^s} j_\tau^s \xrightarrow{L^a} j_\tau^a,$$

whose head and tail satisfies $j_0^a \in A$ and $j_\tau^a \in B$.

Given a sequence of active link sets $\{L_t^s\}_{t=0}^\infty$ and under fixed global policy $\theta$, we say the information at set $A \subseteq \mathcal{N}$ spread to another set $B \subseteq \mathcal{N}$ in $\tau$ time steps (denoted by $I(A) \xrightarrow{\tau} I(B)$) if there exists $(s, a)$ and $(s', a')$ such that $(s_{\mathcal{N} \backslash A}, a_{\mathcal{N} \backslash A}) = (s'_{\mathcal{N} \backslash A}, a'_{\mathcal{N} \backslash A})$ and the distribution of $(s_B(\tau), a_B(\tau))$ given $(s(0), a(0)) = (s, a)$ is different with that given $(s(0), a(0)) = (s', a')$.

We show by induction that $I(A) \xrightarrow{\tau} I(B)$ happens only if $A \xrightarrow{\tau} B$ happens.

If $\tau = 0$, since $I(A) \xrightarrow{0} I(B)$, we see that $A \cap B \neq \emptyset$. Therefore, we can let $j_0^a$ be any agent in $A \cap B$. Hence we also have $A \xrightarrow{0} B$.

Suppose the statement holds for $\tau = t$. When $\tau = t + 1$, suppose that $I(A) \xrightarrow{t+1} I(B)$. Define sets

$$B' := \{j \in \mathcal{N} \mid \exists k \in B, s.t. j \xrightarrow{L^a} k\}, B'' := \{j \in \mathcal{N} \mid \exists k \in B', s.t. j \xrightarrow{L_t^s} k\}.$$

Notice that $B \subseteq B' \subseteq B''$. By the definition of transition probability and policy dependence, we know that the distribution of $a_B(t + 1)$ is decided by $s_{B'}(t + 1)$, and the distribution of $s_{B'}(t + 1)$ is decided by $(s_{B''}(t), a_{B''}(t))$. Therefore, we must have $I(A) \xrightarrow{t} I(B'')$. By the induction hypothesis, we have $A \xrightarrow{t} B''$, which further implies $A \xrightarrow{t+1} B$. This finishes the induction.

Given a sequence of active link sets $\{(L_t^s, L_t^r)\}$, we use $\pi_{t,i}$ to denote the distribution of $\left( s_{N_i(L_t^r)}(t), a_{N_i(L_t^r)}(t) \right)$ given that $(s(0), a(0)) = (s, a)$; we use $\pi'_{t,i}$ to denote the distribution of $\left( s_{N_i(L_t^r)}(t), a_{N_i(L_t^r)}(t) \right)$ given that $(s(0), a(0)) = (s', a')$. We notice that $\pi_{t,i} \neq \pi'_{t,i}$ happens only

if $I(N^{\kappa}_{-i}) \xrightarrow{t} I(N_i(L^r_t))$, which is true only if $N^{\kappa}_{-i} \xrightarrow{t} N_i(L^r_t)$. Recall that $X_i(\kappa)$ is defined as the smallest $t$ such that $N^{\kappa}_{-i} \xrightarrow{t} N_i(L^r_t)$ holds. Hence, we obtain that

$$
\left| Q^{\theta}_i(s, a) - Q^{\theta}_i(s', a') \right|
$$

$$
\leq \mathbb{E}_{\{(L^s_t, L^r_t)\}} \sum_{t=0}^{\infty} \left| \gamma^t \mathbb{E}_{\pi_{t,i}} r_i(s_{N_i(L^r_t)}, a_{N_i(L^r_t)}) - \gamma^t \mathbb{E}_{\pi'_{t,i}} r_i(s_{N_i(L^r_t)}, a_{N_i(L^r_t)}) \right|
$$

$$
\leq \mathbb{E}_{\{(L^s_t, L^r_t)\}} \sum_{t=X_i(\kappa)}^{\infty} \left| \gamma^t \mathbb{E}_{\pi_{t,i}} r_i(s_{N_i(L^r_t)}, a_{N_i(L^r_t)}) - \gamma^t \mathbb{E}_{\pi'_{t,i}} r_i(s_{N_i(L^r_t)}, a_{N_i(L^r_t)}) \right|
$$

$$
\leq \frac{1}{1-\gamma} \mathbb{E}\left[ \gamma^{X_i(\kappa)} \right],
$$

where we use the definition of $X_i(\kappa)$ in the second step.

## B.2 Proof of Corollary 2.2

Given a sequence of active link sets $\{(L^s_t, L^r_t)\}$, let $t = X_i(\kappa)$. By the definition of $X_i(\kappa)$, we assume that a chain of agents

$$
j^a_0 \xrightarrow{L^s_0} j^s_1 \xrightarrow{L^a} j^a_1 \xrightarrow{L^s_1} \cdots \xrightarrow{L^s_{t-1}} j^s_t \xrightarrow{L^a} j^a_t
$$

satisfies $j^a_0 \in N^{\kappa}_{-i}$ and $j^a_t \xrightarrow{L^r_t} i$.

By the triangle inequality and the assumptions of Lemma 2.2, we obtain that

$$
d_{\mathcal{G}}(j^a_0, i) \leq \sum_{\tau=0}^{t-1} \left( d_{\mathcal{G}}(j^a_\tau, j^s_{\tau+1}) + d_{\mathcal{G}}(j^s_{\tau+1}, j^a_{\tau+1}) \right) + d_{\mathcal{G}}(j^a_t, i)
$$

$$
\leq t(\beta + \alpha_1) + \alpha_2.
$$

Therefore, we see that $t$ is lower bounded by $\frac{\kappa - \alpha_2}{\beta + \alpha_1}$, which also gives a lower bound of $X_i(\kappa)$.

## B.3 Proof of Theorem 2.3

To simplify notation, we adopt the same notations as in the proof of Theorem 2.1 (Appendix B.1). Specifically, recall that we use $A \xrightarrow{\tau} B$ to denote the event that there exists a chain

$$
j^a_0 \xrightarrow{L^s_0} j^s_1 \xrightarrow{L^a} j^a_1 \xrightarrow{L^s_1} \cdots \xrightarrow{L^s_{\tau-1}} j^s_\tau \xrightarrow{L^a} j^a_\tau,
$$

whose head and tail satisfies $j^a_0 \in A$ and $j^a_\tau \in B$. We will use $\partial N^{\kappa}_i$ to denote the set of neighbors whose distance to $i$ is $\kappa$, i.e., $\partial N^{\kappa}_i := \{j \in \mathcal{N} \mid d_{\mathcal{G}}(i, j) = \kappa\} = N^{\kappa}_i \setminus N^{\kappa-1}_i$. Define $a_{\kappa} := \mathbb{E}\left[ \gamma^{X_i(\kappa-1)} \right]$. Define function $cat$ (concatenation) such that for a pair of active link sets $(L^s, L^a)$, $(x, y) \in cat(L^s, L^a)$ if and only if $\exists z \in \mathcal{N}$ such that $x \xrightarrow{L^s} z \xrightarrow{L^a} y$.

Before proving Theorem 2.3, we first give an upper bound for the sum of an infinite sequence $\{poly(k+i) \cdot \nu^i\}_{i \in \mathbb{N}}$, where $\nu < 1$ is a positive constant. This result is helpful for showing an upper bound of $P(N^{\kappa}_{-i} \to N^j_i)$.

**Lemma B.1.** *If $m \in \mathbb{N}^*$ and $0 < \nu < 1$ are constants, for all $k \geq \frac{2m}{\ln(1/\nu)}$, we have*

$$
\sum_{i=0}^{\infty} (k+i)^m \nu^i \leq \frac{1}{1 - \sqrt{\nu}} \cdot k^m.
$$

*Proof of Lemma B.1.* Define function $f : \mathbb{R}^+ \cup \{0\} \to \mathbb{R}^+$ as

$$
f(t) = (k+t)^m \cdot \nu^{t/2}.
$$

The derivative of function $f$ is given by

$$f'(t) = (k+t)^{m-1} \cdot \nu^{t/2} \left( m + \frac{1}{2} \ln \nu \cdot (k+t) \right).$$

Since $k \geq \frac{2m}{\ln(1/\nu)}$, $f'(t) \leq 0$ holds for all $t \geq 0$, hence we have $f(t) \leq f(0) = k^m$.

Therefore, we obtain that

$$\sum_{i=0}^{\infty} (k+i)^m \nu^i \leq \sum_{i=0}^{\infty} f(i) \cdot \nu^{i/2}$$

$$\leq k^m \sum_{i=0}^{\infty} \nu^{i/2}$$

$$\leq \frac{1}{1-\sqrt{\nu}} \cdot k^m.$$

$\square$

Now we come back to the proof of Theorem 2.3.

By union bound, we derive an upper bound of the probability that a link $(x,y)$ is in $cat(L^s, L^a)$. Suppose $d \in \mathbb{N}$ is constant that satisfies $d_{\mathcal{G}}(x,y) \geq d$, and the probability $P$ is taken over $(L^s, L^r) \sim \mathcal{D}$:

$$P((x,y) \in cat(L^s, L^a)) = P(\exists z \in \mathcal{N}, (x,z) \in L^s \wedge (z,y) \in L^a)$$

$$\leq \sum_{z: d_{\mathcal{G}}(z,y) \leq \beta} P((x,z) \in L^s)$$

$$\leq c_0 (\beta+1)^{n_0+1} \cdot c\lambda^{d-\beta}$$

$$= c_g \lambda^d, \tag{7}$$

where constant $c_g$ is defined as $c_0 c(\beta+1)^{n_0+1} \lambda^{-\beta}$.

By the assumption on the size of $\kappa$-hop neighborhood, we know that for some constant $c_0$ and $n_0 \in \mathbb{N}^*$, $|\partial N_i^\kappa| \leq c_0(\kappa+1)^{n_0}$ holds for all $\kappa \geq 1$. Let $n_1 := 2n_0$. With the help of Lemma B.1, we show that for some constant $c_2 > 0$, $P\left( N_{-i}^{\kappa-1} \xrightarrow{1} \partial N_i^j \right)$ is upper bounded by $c_2(\kappa+1)^{n_1} \lambda^{\kappa-j}$ for all $j \leq \kappa - 1$ when $\kappa \geq \frac{2n_0}{\ln(1/\lambda)}$:

$$P\left( N_{-i}^{\kappa-1} \xrightarrow{1} \partial N_i^j \right) \leq P\left( \exists x \in N_{-i}^{\kappa-1}, y \in \partial N_i^j \text{ s.t. } (x,y) \in cat(L^s, L^a) \right) \tag{8a}$$

$$\leq \sum_{q=0}^{\infty} P\left( \exists x \in \partial N_i^{\kappa+q}, y \in \partial N_i^j \text{ s.t. } (x,y) \in cat(L^s, L^a) \right) \tag{8b}$$

$$\leq \sum_{q=0}^{\infty} \sum_{x \in \partial N_i^{\kappa+q}, y \in \partial N_i^j} P((x,y) \in cat(L^s, L^a)) \tag{8c}$$

$$\leq \sum_{q=0}^{\infty} \sum_{x \in \partial N_i^{\kappa+q}, y \in \partial N_i^j} c_g \lambda^{(\kappa+q-j)} \tag{8d}$$

$$\leq c_g \lambda^{\kappa-j} \sum_{q=0}^{\infty} \left| \partial N_i^{\kappa+q} \right| \cdot \left| \partial N_i^j \right| \cdot \lambda^q$$

$$\leq c_g c_0^2 (\kappa+1)^{n_0} \lambda^{\kappa-j} \sum_{q=0}^{\infty} (\kappa+q+1)^{n_0} \lambda^q \tag{8e}$$

$$\leq c_2 (\kappa+1)^{n_1} \lambda^{\kappa-j}, \tag{8f}$$

where we use the definition of $N_{-i}^{\kappa-1} \xrightarrow{1} \partial N_i^j$ in (8a); we use union bound in (8b) and (8c); we use the fact that $d_{\mathcal{G}}(x,y) \geq \kappa + q - j, \forall x \in \partial N_i^{\kappa+q}, y \in \partial N_i^j$ and (7) in (8d); we use the bounds

$\left|\partial N_i^j\right| \leq c_0 j^{n_0} \leq c_0 \kappa^{n_0}$ and $\left|\partial N_i^{\kappa+q}\right| \leq c_0(\kappa+q)^{n_0}$ in (8e); we define $c_2 := \frac{c_g c_0^2}{1-\sqrt{\lambda}}$ and use Lemma B.1 in (8f).

Let constants $c_3$ and $q$ be defined as

$$c_3 := \frac{1}{2} \sqrt[4]{\lambda}(1 - \sqrt{\lambda})\left(\frac{1}{\sqrt{\gamma}} - 1\right),$$

$$q := \frac{1}{\ln(1/\lambda)} \max\{(\ln c_2 - \ln c_3 - 2\ln(1 - \sqrt{\gamma})), (2n_1 + 4)\},$$

and define function $p(\kappa) := [q(1 + \ln(\kappa + 1))] + 1$. We can find $\kappa_0 \in \mathbb{Z}^+$ such that $p(\kappa) \geq \kappa$ for all $\kappa \leq \kappa_0$, and $p(\kappa) > \kappa$ for all $\kappa > \kappa_0$.

Let $\rho$ be a constant such that $1 > \rho > \max\{\gamma^{1/(2q)}, \sqrt[4]{\lambda}\}$. Let $C := \rho^{-\max\{q+1, \frac{2n_0}{\ln(1/\lambda)}\}}$. Recall that we define $a_\kappa := \mathbb{E}\left[\gamma^{X_i(\kappa-1)}\right]$, where $X_i(\kappa - 1)$ denotes the smallest $t$ such that $N_{-i}^{\kappa-1} \xrightarrow{t} N_i(L_t^r)$ holds. Now we show by induction that

$$a_\kappa \leq C\rho^{\kappa/(1+\ln(\kappa+1))}, \forall \kappa \geq 1. \tag{9}$$

Since $a_\kappa \leq 1$, (9) clearly holds when $\kappa \leq \kappa_0$. To see this, recall that we have $\kappa \leq p(\kappa)$ and $C \geq \rho^{-(q+1)}$ by definition, thus the right hand side of (9) can be lower bounded by

$$C\rho^{\kappa/(1+\ln(\kappa+1))} \geq \rho^{-(q+1)} \cdot \rho^{p(\kappa)/(1+\ln(\kappa+1))} \geq \rho^{-(q+1)} \cdot \rho^{q+1} = 1.$$

When $\kappa > \kappa_0$, we have $\kappa > p(\kappa)$. Recall that $a_\kappa := \mathbb{E}\left[\gamma^{X_i(\kappa-1)}\right]$. Notice that $X_i(\kappa - 1) = 0$ if and only if $N_{-i}^{\kappa-1} \cap N_i(L_0^r) \neq \emptyset$. To simplify the notation, we denote the event $N_{-i}^{\kappa-1} \cap N_i(L_0^r) \neq \emptyset$ by $E_0$. Using this and the idea of dynamic programming, we see that

$$a_\kappa \leq \gamma \left(P\{\left(\neg N_{-i}^{\kappa-1} \xrightarrow{1} N_i^{\kappa-1}\right) \wedge \neg E_0\}a_\kappa + \sum_{j=0}^{\kappa-1} P\{\left(N_{-i}^\kappa \xrightarrow{1} \partial N_i^j\right) \wedge \left(\neg N_{-i}^\kappa \xrightarrow{1} N_i^{j-1}\right) \wedge \neg E_0\}a_j\right)$$

$$+ P(E_0)$$

$$\leq \gamma \left(P\{\neg N_{-i}^{\kappa-1} \xrightarrow{1} N_i^{\kappa-1}\}a_\kappa + \sum_{j=0}^{\kappa-1} P\{\left(N_{-i}^\kappa \xrightarrow{1} \partial N_i^j\right) \wedge \left(\neg N_{-i}^\kappa \xrightarrow{1} N_i^{j-1}\right)\}a_j\right) + P(E_0), \tag{10}$$

where the probability $P$ are taken over $(L_0^s, L_0^r) \sim D$.

Since $\kappa \geq p(\kappa) \geq q \geq \frac{2n_1}{\ln(1/\lambda)} \geq \frac{2n_0}{\ln(1/\lambda)}$, by Lemma B.1, we see that

$$P(E_0) = P\{\exists j \in N_{-i}^{\kappa-1} \text{ s.t. } (j, i) \in L^r\} \leq \sum_{q=0}^{\infty} cc_0(\kappa+q+1)^{n_0}\lambda^{\kappa+q} \leq \frac{cc_0}{1-\sqrt{\lambda}}(\kappa+1)^{n_0+1}\lambda^\kappa.$$

Substituting this into (10) and rearranging the terms gives

$$\left(1 - \gamma P\{\neg N_{-i}^{\kappa-1} \xrightarrow{1} N_i^{\kappa-1}\}\right)a_\kappa \leq \gamma \sum_{j=\kappa-p(\kappa)+1}^{\kappa-1} P\{\left(N_{-i}^{\kappa-1} \xrightarrow{1} \partial N_i^j\right) \wedge \left(\neg N_{-i}^{\kappa-1} \xrightarrow{1} N_i^{j-1}\right)\}a_j$$

$$+ \gamma \sum_{j=0}^{\kappa-p(\kappa)} P\{\left(N_{-i}^{\kappa-1} \xrightarrow{1} \partial N_i^j\right) \wedge \left(\neg N_{-i}^{\kappa-1} \xrightarrow{1} N_i^{j-1}\right)\}a_j$$

$$+ \frac{cc_0}{1-\sqrt{\lambda}}(\kappa+1)^{n_0+1}\lambda^\kappa. \tag{11}$$

For simplicity, we define $\rho_\kappa := \rho^{1/(1+\ln(\kappa+1))}$. By the induction assumption, we have that

$$a_j \leq C\rho^{j/(\ln(j+1)+1)} \leq C\rho^{j/(\ln(\kappa+1)+1)} = C\rho_\kappa^j.$$

Substituting this into (11) gives that

$$\left(1 - \gamma P\{\neg N_{-i}^{\kappa-1} \xrightarrow{1} N_i^{\kappa-1}\}\right)a_\kappa \le C\gamma \sum_{j=\kappa-p(\kappa)+1}^{\kappa-1} P\{\left(N_{-i}^{\kappa-1} \xrightarrow{1} \partial N_i^j\right) \wedge \left(\neg N_{-i}^{\kappa-1} \xrightarrow{1} N_i^{j-1}\right)\}\rho_\kappa^j$$

$$+ C\gamma \sum_{j=0}^{\kappa-p(\kappa)} P\{\left(N_{-i}^{\kappa-1} \xrightarrow{1} \partial N_i^j\right) \wedge \left(\neg N_{-i}^{\kappa-1} \xrightarrow{1} N_i^{j-1}\right)\}\rho_\kappa^j$$

$$+ \frac{c_0}{1 - \sqrt{\lambda}}(\kappa+1)^{n_0+1}\lambda^\kappa. \tag{12}$$

By the definition of $p(\kappa)$ and $q$, we see that

$$\lambda^{-p(\kappa)} \ge \lambda^{-q(1+\ln(\kappa+1))} = \lambda^{-q} \cdot (\kappa+1)^{q\ln(1/\lambda)} \ge \frac{c_2}{c_3(1-\sqrt{\gamma})^2} \cdot (\kappa+1)^{n_1} \ge \frac{c_2}{c_3(1-\gamma)} \cdot (\kappa+1)^{n_1}.$$

Therefore, we obtain the upper bound

$$P\{\left(N_{-i}^{\kappa-1} \xrightarrow{1} \partial N_i^j\right) \wedge \left(\neg N_{-i}^\kappa \xrightarrow{1} N_i^{j-1}\right)\} \le P\{N_{-i}^{\kappa-1} \xrightarrow{1} \partial N_i^j\}$$

$$\le c_2(\kappa+1)^{n_1}\lambda^{(\kappa-j)}$$

$$\le (1-\gamma)c_3\lambda^{(\kappa-p(\kappa)-j)}.$$

Using this and divide both sides of (12) by $\left(1 - \gamma P\{\neg N_{-i}^\kappa \xrightarrow{1} N_i^{\kappa-1}\}\right)$, we see that

$$a_\kappa \le \gamma\left(C\rho_\kappa^{\kappa-p(\kappa)+1} + Cc_3(\rho_\kappa^{\kappa-p(\kappa)} + \lambda^1 \cdot \rho_\kappa^{\kappa-p(\kappa)-1} + \lambda^2 \cdot \rho_\kappa^{\kappa-p(\kappa)-2} + \cdots)\right)$$

$$+ \frac{c_0}{(1-\gamma)(1-\sqrt{\lambda})}(\kappa+1)^{n_0+1}\lambda^\kappa, \tag{13}$$

where we also use the fact that

$$\sum_{j=\kappa-p+1}^{\kappa-1} P\{\left(N_{-i}^{\kappa-1} \xrightarrow{1} \partial N_i^j\right) \wedge \left(\neg N_{-i}^{\kappa-1} \xrightarrow{1} N_i^{j-1}\right)\} \le 1 - \gamma P\{\neg N_{-i}^{\kappa-1} \xrightarrow{1} N_i^{\kappa-1}\}.$$

By the definition of $p(\kappa), q$ and $c_2$, we have that

$$\lambda^{\frac{\kappa}{4}} \le \lambda^{\frac{p(\kappa)}{4}} \le (\kappa+1)^{-\frac{q\ln(1/\lambda)}{4}} \le (\kappa+1)^{-n_0-1}$$

and

$$\lambda^{\frac{\kappa}{2}} \le \lambda^{\frac{p(\kappa)}{2}} \le \lambda^{\frac{q}{2}} \le \frac{(1-\sqrt{\gamma})(1-\gamma)(1-\sqrt{\lambda})}{2c_0},$$

which implies

$$\lambda^{\frac{3\kappa}{4}} \le \frac{(1-\sqrt{\gamma})(1-\gamma)(1-\sqrt{\lambda})}{2c_0(\kappa+1)^{n_0+1}}. \tag{14}$$

Dividing both sides of (13) by $C\rho_\kappa^\kappa$ gives that

$$\frac{a_\kappa}{C\rho_\kappa^\kappa} \le \gamma\left(\frac{1}{\rho_\kappa^{p(\kappa)-1}} + \frac{c_3}{\rho_\kappa^{p(\kappa)}} \cdot \frac{1}{1-(\lambda/\rho_\kappa)}\right) + \frac{c_0}{(1-\gamma)(1-\sqrt{\lambda})}(\kappa+1)^{n_0+1}\lambda^{\frac{3\kappa}{4}} \tag{15a}$$

$$\le \gamma\left(\frac{1}{\rho^q} + \frac{1}{\rho^{q+1}} \cdot \frac{c_3}{1-\sqrt{\lambda}}\right) + \frac{1}{2}(1-\sqrt{\gamma}) \tag{15b}$$

$$= \frac{\gamma}{\rho^q}\left(1 + \frac{c_3}{\rho(1-\sqrt{\lambda})}\right) + \frac{1}{2}(1-\sqrt{\gamma})$$

$$\le \sqrt{\gamma} \cdot \frac{1}{2}\left(1 + \frac{1}{\sqrt{\gamma}}\right) + \frac{1}{2}(1-\sqrt{\gamma}) \tag{15c}$$

$$= 1,$$

where we use $\rho_\kappa = \rho^{1/(1+\ln\kappa)} \ge \rho \ge \sqrt[4]{\lambda}$ in (15a); we use $\rho_\kappa \ge \sqrt[4]{\lambda}$, $p = \lceil q(1 + \ln\kappa)\rceil + 1$, and (14) in (15b); we use $c_3 = \sqrt{\lambda}(1-\sqrt{\lambda})(\sqrt{\gamma}-1) \le \rho(1-\sqrt{\lambda})(\sqrt{\gamma}-1)$ and $\rho \ge \gamma^{1/(2q)}$ in (15c).

## B.4 Proof of Theorem 2.4

In the Critic part of Algorithm 1, since the policy is fixed to be $\theta(m)$, the pair $(s, a)$ can be viewed as the state of a Markov chain $\mathcal{C}$, and $Q^{\theta(m)}(s, a)$ in the original MDP corresponds to the value function $V^*((s, a))$ on $\mathcal{C}$. Define the state aggregation map $h$ such that $h((s, a)) = (s_{N_i^\kappa}, a_{N_i^\kappa})$. By the $\mu$-decay property, we see that if $h((s, a)) = h((s', a'))$, then

$$|V^*((s, a)) - V^*((s', a'))| = \left| Q^{\theta(m)}(s, a) - Q^{\theta(m)}(s', a') \right| \leq \mu(\kappa).$$

Note that Assumption 2.1 implies that Assumption 3.1 holds for $\mathcal{C}$. Thus, we can apply Theorem 3.2 to finish the proof of Theorem 2.4.

## B.5 Proof of Theorem 2.5

Before showing Theorem 2.5, we first state a theorem concerning the actor part of Algorithm 1. The proof is deferred to Appendix B.6.

**Theorem B.2.** *Under the same assumption as Theorem 2.5, suppose inner loop length $T$ is sufficiently large such that $T+1 \geq \log_\gamma((1-\gamma)\mu(\kappa))$ and with probability at least $1-\frac{\delta}{2}$, the following inequality holds for all agents $i \in \mathcal{N}$:*

$$\sup_{m \leq M-1} \sup_{(s,a) \in \mathcal{S} \times \mathcal{A}} \left| Q_i^{\theta(m)}(s, a) - \hat{Q}^T(s_{N_i^\kappa}, a_{N_i^\kappa}) \right| \leq \frac{\iota \mu(\kappa)}{1-\gamma},$$

*where $\iota$ is a positive constant. Suppose the actor step size satisfies $\eta_m = \frac{\eta}{\sqrt{m+1}}$ with $\eta \leq \frac{1}{4W'}$. Define $C_M := \frac{2}{\eta(1-\gamma)} + \frac{8W^2\sqrt{\log M \log \frac{4}{\delta}} + 96W'W^2\eta \log M}{(1-\gamma)^4}$. Then, with probability at least $1 - \delta$,*

$$\frac{\sum_{m=0}^{M-1} \eta_m \|\nabla J(\theta(m))\|^2}{\sum_{m=0}^{M-1} \eta_m} \leq \frac{C_M}{\sqrt{M+1}} + \frac{2(2+\iota)W^2\mu(\kappa)}{(1-\gamma)^4}. \tag{16}$$

As a remark, note that the left hand side of (16) is a weighted average of the squared norm of the gradients $\nabla J(\theta(m))$. We say the algorithm has reached an $O(\epsilon)$-approximate stationary point if the left hand side of (16) is in the order of $O(\epsilon)$.

Now we come back to the proof of Theorem 2.5. Let constant $\iota = 2$ in Theorem B.2. By Theorem B.2, to satisfy

$$\frac{\sum_{m=0}^{M-1} \eta_m \|\nabla J(\theta(m))\|^2}{\sum_{m=0}^{M-1} \eta_m} \leq O(\epsilon),$$

it suffices to guarantee that

$$\frac{C_M}{\sqrt{M+1}} = O(\epsilon), \text{ and } \frac{2(2+\iota)W^2\mu(\kappa)}{(1-\gamma)^4} = O(\epsilon).$$

These can be satisfied by letting

$$M = \tilde{\Omega}\left(\epsilon^{-2}\left(\frac{(W')^2}{(1-\gamma)^2} + \frac{W^4(1+\log(1/\delta))}{(1-\gamma)^8}\right)\right), \mu(\kappa) = O\left(W^{-2}(1-\gamma)^4\epsilon\right).$$

To satisfy

$$\sup_{m \leq M-1} \sup_{(s,a) \in \mathcal{S} \times \mathcal{A}} \left| Q_i^{\theta(m)}(s, a) - \hat{Q}^T(s_{N_i^\kappa}, a_{N_i^\kappa}) \right| \leq \frac{\iota \mu(\kappa)}{1-\gamma},$$

with probability at least $1 - \frac{\delta}{2}$, by Corollary 2.4, it suffices to select $T$ such that

$$\frac{1}{\sqrt{T+t_0}} \cdot \frac{40H}{(1-\gamma)^2} \sqrt{K_2 \log T \left(\log\left(\frac{4f(\kappa)K_2 T}{\delta}\right) + \log\log T\right)}$$

$$+ \frac{1}{T+t_0} \cdot \frac{8}{(1-\gamma)^2} \max\left\{\frac{144K_2 H \log T}{\sigma'(\kappa)} + C_3, 2K_2 \log T + t_0\right\}$$

$$\leq \frac{\mu(\kappa)}{1-\gamma}.$$

Recall that

$$\mu(\kappa) = O\big(W^{-2}(1-\gamma)^4 \epsilon\big), H \geq \frac{2}{(1-\gamma)\sigma'(\kappa)}, t_0 = \max(4H, 2K_2 \log T).$$

Hence the required number of inner loop is

$$T = \tilde{\Omega}\bigg(\frac{W^4(K_2(\log f(\kappa) + \log(1/\delta) + 1) + K_1)}{\epsilon^2(1-\gamma)^{12}\sigma'(\kappa)^2}\bigg).$$

## B.6 Proof of Theorem B.2

While Theorem 5 in [38] studies the error bound of Scalable Actor Critic as a whole, we want to decouple the effect of the inner loop and the outer loop in Theorem B.2. Our proof of Theorem B.2 uses similar techniques with the proof in [38], but we extend the analysis to a more general dependence model.

According to Algorithm 1, at iteration $m$, agent $i$ performs gradient ascent by

$$\theta_i(m+1) = \theta_i(m) + \eta_m \hat{g}_i(m),$$

with step size $\eta_m = \frac{\eta}{\sqrt{m+1}}$. The approximate local gradient $\hat{g}_i(m)$ is given by

$$\hat{g}_i(m) = \sum_{t=0}^T \gamma^t \frac{1}{n} \sum_{j \in N_i^\kappa} \hat{Q}_j^{m,T}\Big(s_{N_j^\kappa}(t), a_{N_j^\kappa}(t)\Big) \nabla_{\theta_i} \log \zeta_i^{\theta_i(m)}\Big(a_i(t) \mid s_{N_i^\beta}(t)\Big).$$

Recall that the true local gradient is given by

$$\nabla_{\theta_i} J(\theta(m)) = \sum_{t=0}^\infty \mathbb{E}_{s \sim \pi_t^{\theta(m)}, a \sim \zeta^{\theta(m)}(\cdot|s)} \gamma^t Q^{\theta(m)}(s,a) \nabla_{\theta_i} \log \zeta_i^{\theta_i(m)}\Big(a_i(t) \mid s_{N_i^\beta}(t)\Big),$$

where we use $\pi_t^\theta$ to denote the distribution of global state $s(t)$ under fixed policy $\theta$.

To bound $\|\hat{g}(m) - \nabla_\theta J(\theta(m))\|$, we define intermediate quantities $g(m)$ and $h(m)$ whose $i$'th component is given by

$$g_i(m) = \sum_{t=0}^T \gamma^t \frac{1}{n} \sum_{j \in N_i^\kappa} Q_j^{\theta(m)}(s(t), a(t)) \nabla_{\theta_i} \log \zeta_i^{\theta_i(m)}\Big(a_i(t) \mid s_{N_i^\beta}(t)\Big),$$

$$h_i(m) = \sum_{t=0}^T \mathbb{E}_{s \sim \pi_t^{\theta(m)}, a \sim \zeta^{\theta(m)}(\cdot|s)} \gamma^t \frac{1}{n} \sum_{j \in N_i^\kappa} Q_j^{\theta(m)}(s,a) \nabla_{\theta_i} \log \zeta_i^{\theta_i(m)}\Big(a_i(t) \mid s_{N_i^\beta}(t)\Big).$$

**Lemma B.3.** *We have almost surely, $\forall m \leq M$,*

$$\max(\|\hat{g}(m)\|, \|g(m)\|, \|h(m)\|, \|\nabla J(\theta(m))\|) \leq \frac{W}{(1-\gamma)^2}.$$

To show Lemma B.3, we only need to replace $\zeta_i^{\theta_i(m)}(a_i(t) \mid s_i(t))$ by $\zeta_i^{\theta_i(m)}\Big(a_i(t) \mid s_{N_i^\beta}(t)\Big)$ in the proof of Lemma 17 in [38].

Notice that

$$\hat{g}(m) - \nabla J(\theta(m)) = e^1(m) + e^2(m) + e^3(m),$$

where

$$e^1(m) := \hat{g}(m) - g(m), e^2(m) := g(m) - h(m), e^3(m) := h(m) - \nabla J(\theta(m)).$$

To bound $\|\hat{g}(m) - \nabla J(\theta(m))\|$, we only need to bound $e_1(m), e_2(m), e_3(m)$ separately.

**Lemma B.4.** *With probability at least $1 - \frac{\delta}{2}$, we have*

$$\sup_{0 \leq m \leq M-1} \|e^1(m)\| \leq \frac{\iota W \mu(\kappa)}{(1-\gamma)^2}.$$

*Proof of Lemma B.4.* By the assumption that

$$\sup_{m \leq M-1} \sup_{i \in \mathcal{N}} \sup_{(s,a) \in \mathcal{S} \times \mathcal{A}} \left| Q_i^{\theta(m)}(s,a) - \hat{Q}^T(s_{N_i^\kappa}, a_{N_i^\kappa}) \right| \leq \frac{\iota \cdot \mu(\kappa)}{1-\gamma},$$

we have for all $m \leq M-1$ and $i \in \mathcal{N}$,

$$
\begin{aligned}
&\|\hat{g}_i(m) - g_i(m)\| \\
&\leq \left\| \sum_{t=0}^{T} \gamma^t \frac{1}{n} \sum_{j \in N_i^\kappa} \left[ \hat{Q}_j^{m,T}\left(s_{N_j^\kappa}(t), a_{N_j^\kappa}(t)\right) - Q_j^{\theta(m)}(s(t), a(t)) \right] \nabla_{\theta_i} \log \zeta_i^{\theta_i(m)}\left(a_i(t) \mid s_{N_i^\beta}(t)\right) \right\| \\
&\leq \sum_{t=0}^{T} \gamma^t \frac{1}{n} \sum_{j \in N_i^\kappa} \left| \hat{Q}_j^{m,T}\left(s_{N_j^\kappa}(t), a_{N_j^\kappa}(t)\right) - Q_j^{\theta(m)}(s(t), a(t)) \right| \left\| \nabla_{\theta_i} \log \zeta_i^{\theta_i(m)}\left(a_i(t) \mid s_{N_i^\beta}(t)\right) \right\| \\
&\leq \sum_{t=0}^{T} \gamma^t \frac{\iota \cdot \mu(\kappa)}{1-\gamma} W_i \\
&< \frac{2\iota W_i \cdot \mu(\kappa)}{(1-\gamma)^2}.
\end{aligned}
$$

Combining all $n$ dimensions finishes the proof. $\qquad\square$

**Lemma B.5.** *With probability at least* $1 - \frac{\delta}{2}$*, we have*

$$\left| \sum_{m=0}^{M-1} \eta_m \langle \nabla J(\theta(m)), e^2(m) \rangle \right| \leq \frac{2W^2}{(1-\gamma)^4} \sqrt{2 \sum_{m=0}^{M-1} \eta_m^2 \log \frac{4}{\delta}}.$$

To show Lemma B.5, we only need to replace $\zeta_i^{\theta_i(m)}(a_i(t) \mid s_i(t))$ by $\zeta_i^{\theta_i(m)}\left(a_i(t) \mid s_{N_i^\beta}(t)\right)$ in the proof of Lemma 19 in [38].

**Lemma B.6.** *When* $T + 1 \geq \log_\gamma((1-\gamma)\mu(\kappa))$*, we have almost surely*

$$\left\| e^3(m) \right\| \leq \frac{2W\mu(\kappa)}{1-\gamma}.$$

To show Lemma B.6, we only need to replace $\zeta_i^{\theta_i(m)}(a_i(t) \mid s_i(t))$ with $\zeta_i^{\theta_i(m)}\left(a_i(t) \mid s_{N_i^\beta}(t)\right)$ and replace $c\rho^{\kappa+1}$ with $\mu(\kappa)$ in the proof of Lemma 20 in [38].

Now we come back to the proof of Theorem B.2. Using the identical steps with the proof of Theorem 5 in [38], we can obtain that (equation (44) in [38])

$$\sum_{m=0}^{M-1} \frac{1}{2} \eta_m \|\nabla J(\theta(m))\|^2 \leq J(\theta(m)) - J(\theta(0)) - \sum_{m=0}^{M-1} \eta_m \epsilon_{m,0} + \sum_{m=0}^{M-1} \eta_m \epsilon_{m,1} + \sum_{m=0}^{M-1} \eta_m^2 \epsilon_{m,2}, \quad (17)$$

where

$$
\begin{aligned}
\epsilon_{m,0} &= \langle \nabla J(\theta(m)), e^2(m) \rangle, \\
\epsilon_{m,1} &= \|\nabla J(\theta(m))\|(\|e^1(m)\| + \|e^3(m)\|), \\
\epsilon_{m,2} &= 2W'(\|e^1(m)\|^2 + \|e^2(m)\|^2 + \|e^3(m)\|^2).
\end{aligned}
$$

By Lemma B.5, we have with probability at least $1 - \frac{\delta}{2}$,

$$\left| \sum_{m=0}^{M-1} \eta_m \epsilon_{m,0} \right| \leq \frac{2W^2}{(1-\gamma)^4} \sqrt{2 \sum_{m=0}^{M-1} \eta_m^2 \log \frac{4}{\delta}}. \qquad (18)$$

By Lemma B.4 and Lemma B.6, we have with probability at least $1 - \frac{\delta}{2}$,

$$
\begin{aligned}
\sup_{m \leq M-1} \epsilon_{m,1} &\leq \frac{W}{(1-\gamma)^2} \left( \sup_{m \leq M-1} \left\| e^1(m) \right\| + \sup_{m \leq M-1} \left\| e^3(m) \right\| \right) \\
&\leq \frac{(2+\iota)W^2 \mu(\kappa)}{(1-\gamma)^4}.
\end{aligned}
\tag{19}
$$

By Lemma B.3, we have almost surely $\max(\left\| e^1(m) \right\|, \left\| e^2(m) \right\|, \left\| e^3(m) \right\|) \leq 2\frac{W}{(1-\gamma)^2}$, and hence almost surely

$$
\begin{aligned}
\sup_{m \leq M-1} \epsilon_{m,2} &= 2W' \left( \left\| e^1(m) \right\|^2 + \left\| e^2(m) \right\|^2 + \left\| e^3(m) \right\|^2 \right) \\
&\leq \frac{24W'W^2}{(1-\gamma)^4}.
\end{aligned}
\tag{20}
$$

By union bound, (18), (19), and (20) hold simultaneously with probability $1 - \delta$. Combining them with (17) gives

$$
\begin{aligned}
&\frac{\sum_{m=0}^{M-1} \eta_m \|\nabla J(\theta(m))\|^2}{2 \sum_{m=0}^{M-1} \eta_m} \\
&\leq \frac{(J(\theta(M)) - J(\theta(0))) + \left| \sum_{m=0}^{M-1} \eta_m \epsilon_{m,0} \right| + \sup_{m \leq M-1} \epsilon_{m,2} \sum_{m=0}^{M-1} \eta_m^2}{\sum_{m=0}^{M-1} \eta_m} + 2 \sup_{m \leq M-1} \epsilon_{m,1}.
\end{aligned}
\tag{21}
$$

We can use identical steps with the proof of Theorem 5 in [38] to bound the first term in (21), and use (19) to bound the second term in (21). This completes the proof.

## C Stochastic Approximation Scheme

### C.1 Contraction of the Update Operator

To show that the equation $\Pi F(\Phi x) = x$ has a unique solution $x^*$, by the Banach–Caccioppoli fixed-point theorem, it suffices to show that operator $\Pi F(\Phi \cdot)$ is a $\gamma$-contraction in $\|\cdot\|_v$.

**Proposition C.1.** *If Assumption 3.2 holds, operator $\Pi F(\Phi \cdot)$ is a contraction in $\|\cdot\|_v$, i.e., for any $x, y \in \mathbb{R}^{\mathcal{M}}$, $\|\Pi F(\Phi x) - \Pi F(\Phi y)\|_v \leq \gamma \|x - y\|_v$.*

To prove this proposition, we first show both operator $\Pi$ and operator $\Phi$ are non-expansive in $\|\cdot\|_v$ before combining them with $F$.

*Proof of Proposition C.1.* We first show that operator $\Pi$ is non-expansive in $\|\cdot\|_v$, i.e. for any $x, y \in \mathbb{R}^{\mathcal{N}}$, we have

$$
\|\Pi x - \Pi y\|_v \leq \|x - y\|_v.
\tag{22}
$$

Since $\Pi$ is a linear operator, it suffices to show that for any $x \in \mathbb{R}^{\mathcal{N}}$, $\|\Pi x\|_v \leq \|x\|_v$.

Recall that $\forall j \in \mathcal{M}, h^{-1}(j) := \{i \in \mathcal{N} \mid h(i) = j\}$. Using this notation, the $j$ th element of vector $\Pi x$ is given by

$$
(\Pi x)_j = \frac{1}{\sum_{i \in h^{-1}(j)} d_i} \left( \Phi^\top D x \right)_j = \frac{1}{\sum_{i \in h^{-1}(j)} d_i} \cdot \sum_{i \in h^{-1}(j)} d_i x_i.
$$

Hence we see that

$$
\frac{\left| (\Pi x)_j \right|}{v_j} \leq \frac{1}{\sum_{i \in h^{-1}(j)} d_i} \cdot \sum_{i \in h^{-1}(j)} d_i \frac{|x_i|}{v_j} \leq \sup_{i \in h^{-1}(j)} \frac{|x_i|}{v_j}.
\tag{23}
$$

By taking $\sup_j$ on both sides of (23), we see that

$$\|\Pi x\|_v = \sup_{j \in \mathcal{M}} \frac{\left|(\Pi x)_j\right|}{v_j} \le \sup_{j \in \mathcal{M}} \sup_{i \in h^{-1}(j)} \frac{|x_i|}{v_j} = \sup_{i \in \mathcal{N}} \frac{|x_i|}{v_{h(i)}} = \|x\|_v, \qquad (24)$$

where we use the definition of $\|\cdot\|_v$ on $\mathbb{R}^{\mathcal{N}}$ in the last equation. Hence we have shown that $\Pi$ is non-expansive in $\|\cdot\|_v$ (inequality (22)).

We can also show that for any $x, y \in \mathbb{R}^{\mathcal{M}}$, we have

$$\|\Phi x - \Phi y\|_v = \|x - y\|_v. \qquad (25)$$

Since $\Phi$ is a linear operator, we only need to show that for any $x \in \mathbb{R}^{\mathcal{M}}$, $\|\Phi x\|_v = \|x\|_v$.

Since $(\Phi x)_i = x_{h(i)}, \forall i \in \mathcal{N}$, by the definition of $\|\cdot\|_v$ on $\mathbb{R}^{\mathcal{N}}$, we see that

$$\|\Phi x\|_v = \sup_{i \in \mathcal{N}} \frac{|(\Phi x)_i|}{v_{h(i)}} = \sup_{i \in \mathcal{N}} \frac{\left|x_{h(i)}\right|}{v_{h(i)}} = \sup_{j \in \mathcal{M}} \frac{|x_j|}{v_j} = \|x\|_v.$$

Hence we have shown that $\Phi$ is non-expansive in $\|\cdot\|_v$ (equation (25)).

Therefore, for any $x, y \in \mathbb{R}^{\mathcal{M}}$, we have

$$\|\Pi F(\Phi x) - \Pi F(\Phi y)\|_v \le \|F(\Phi x) - F(\Phi y)\|_v \qquad (26a)$$
$$\le \gamma \|\Phi x - \Phi y\|_v \qquad (26b)$$
$$= \gamma \|x - y\|_v, \qquad (26c)$$

where we use (22) in (26a); Assumption 3.2 in (26b); (25) in (26c). $\qquad \square$

## C.2   Proof of Theorem 3.1

The proof approach of Theorem 3.1 is similar to the proof of Theorem 4 in [37]. Specifically, we show an upper bound for $\|x(t) - x^*\|_v$ by induction on time step $t$. To do so, we divide the whole proof into three steps: In Step 1, we manipulate the update rule (3) so that it can be written in a recursive form of sequence $\|x(t) - x^*\|_v$ (see Lemma C.1); In Step 2, we bound the effect of noise terms in the recursive form we obtained in Step 1; In Step 3, we combine the first two steps to finish the induction.

For simplicity of notation, we use $e_i$ to denote the indicator vector in $\mathbb{R}^n$, i.e. the $i$ th entry is 1 and all other entries are 0. We also use $\xi_i$ to denote the indicator vector in $\mathbb{R}^m$.

One of the main proof techniques used in [37] is to consider $D_t = \mathbb{E} e_{i_t} e_{i_t}^\top \mid \mathcal{F}_{t-\tau}$, which is the distribution of $i_t$ condition on $\mathcal{F}_{t-\tau}$, in the coefficients of the recursive relationship of sequence $\|x(t) - x^*\|_v$. However, this approach does not work in the more general setting we consider because $x^*$ may not be the stationary point of operator $(\Phi^\top D_t \Phi)^{-1} \phi^\top D_t F(\Phi \cdot)$. As a result, we cannot decompose $\|x(t) - x^*\|_v$ recursively if we use $D_t$ in the coefficients. To overcome this difficulty, we use $D = diag(d_1, \cdots, d_n)$, which is the stationary distribution of $i_t$, in the coefficients of the recursive relationship (Lemma C.1).

Now we begin the technical part of our proof.

**Step 1: Decomposition of Error.** Let $D_t = \mathbb{E} e_{i_t} e_{i_t}^\top \mid \mathcal{F}_{t-\tau}$, where $\tau$ is a parameter that we will tune later. Then $D_t$ is a $\mathcal{F}_{t-\tau}$-measurable $n$-by-$n$ diagonal random matrix, with its $i$'th entry being $d_{t,i} = \mathbb{P}(i_t = i \mid \mathcal{F}_{t-\tau})$. Recall that $D = diag(d_1, \cdots, d_n)$, where $d$ is the stationary distribution of the Markov Chain $\{i_t\}$.

Notice that for all $i \in \mathcal{N}$, we have $\xi_{h(i)} = \Phi^\top e_i$. We can rewrite the update rule as

$$
\begin{aligned}
x(t+1) &= x(t) + \alpha_t[e_{i_t}^\top F(\Phi x(t)) - \xi_{h(i_t)}^\top x(t) + w(t)]\xi_{h(i_t)} \\
&= x(t) + \alpha_t[\xi_{h(i_t)}e_{i_t}^\top F(\Phi x(t)) - \xi_{h(i_t)}\xi_{h(i_t)}^\top x(t) + w(t)\xi_{h(i_t)}] \\
&= x(t) + \alpha_t\Phi^\top\left[e_{i_t}e_{i_t}^\top(F(\Phi x(t)) - \Phi x(t)) + w(t)e_{i_t}\right] \quad (27\text{a}) \\
&= x(t) + \alpha_t\left[\Phi^\top DF(\Phi x(t)) - \Phi^\top D\Phi x(t)\right] \\
&\quad + \alpha_t\Phi^\top\left[(e_{i_t}e_{i_t}^\top - D)(F(\Phi x(t)) - \Phi x(t)) + w(t)e_{i_t}\right] \\
&= x(t) + \alpha_t\left[\Phi^\top DF(\Phi x(t)) - \Phi^\top D\Phi x(t)\right] \\
&\quad + \alpha_t\Phi^\top\left[(e_{i_t}e_{i_t}^\top - D)(F(\Phi x(t-\tau)) - \Phi x(t-\tau)) + w(t)e_{i_t}\right] \\
&\quad + \alpha_t\Phi^\top(e_{i_t}e_{i_t}^\top - D)[F(\Phi x(t)) - F(\Phi x(t-\tau)) - \Phi(x(t) - x(t-\tau))] \\
&= (I - \alpha_t\Phi^\top D\Phi)x(t) + \alpha_t\Phi^\top DF(\Phi x(t)) + \alpha_t(\epsilon(t) + \psi(t)), \quad (27\text{b})
\end{aligned}
$$

where in (27a), we use $\xi_{h(i_t)} = \Phi^\top e_{i_t}$. Additionally, in (27b), we define

$$
\epsilon(t) = \Phi^\top\left[(e_{i_t}e_{i_t}^\top - D)(F(\Phi x(t-\tau)) - \Phi x(t-\tau)) + w(t)e_{i_t}\right]
$$

and

$$
\psi(t) = \Phi^\top(e_{i_t}e_{i_t}^\top - D)[F(\Phi x(t)) - F(\Phi x(t-\tau)) - \Phi(x(t) - x(t-\tau))].
$$

We further decompose $\epsilon(t)$ as $\epsilon(t) = \epsilon_1(t) + \epsilon_2(t)$, where $\epsilon_1(t)$ and $\epsilon_2(t)$ are defined as

$$
\epsilon_1(t) = \Phi^\top\left[(e_{i_t}e_{i_t}^\top - D_t)(F(\Phi x(t-\tau)) - \Phi x(t-\tau)) + w(t)e_{i_t}\right]
$$

and

$$
\epsilon_2(t) = \Phi^\top(D_t - D)(F(\Phi x(t-\tau)) - \Phi x(t-\tau)).
$$

We see that condition on $\mathcal{F}_{t-\tau}$, the expected value of $\epsilon_1(t)$ is zero, i.e.

$$
\begin{aligned}
&\mathbb{E}\epsilon_1(t) \mid \mathcal{F}_{t-\tau} \\
&= \Phi^\top\mathbb{E}\left[(e_{i_t}e_{i_t}^\top - D_t) \mid \mathcal{F}_{t-\tau}\right][F(\Phi x(t-\tau)) - \Phi x(t-\tau)] + \Phi^\top\mathbb{E}[\mathbb{E}[w(t) \mid \mathcal{F}_t]e_{i_t} \mid \mathcal{F}_{t-\tau}] \\
&= 0.
\end{aligned}
$$

Recall that matrix $\Pi$ is defined as

$$
\Pi = \left(\Phi^\top D\Phi\right)^{-1}\Phi^\top D.
$$

By expanding (27) recursively, we obtain that

$$
\begin{aligned}
x(t+1) &= \prod_{k=\tau}^{t}\left(I - \alpha_k\Phi^\top D\Phi\right)x(\tau) + \sum_{k=\tau}^{t}\alpha_k\left(\prod_{l=k+1}^{t}(I - \alpha_l\Phi^\top D\Phi)\right)\Phi^\top DF(\Phi x(k)) \\
&\quad + \sum_{k=\tau}^{t}\alpha_k\left(\prod_{l=k+1}^{t}(I - \alpha_l\Phi^\top D\Phi)\right)(\epsilon(k) + \psi(k)) \\
&= \tilde{B}_{\tau-1,t}x(\tau) + \sum_{k=\tau}^{t}B_{k,t}\Pi F(\Phi x(k)) + \sum_{k=\tau}^{t}\alpha_k\tilde{B}_{k,t}(\epsilon(k) + \psi(k)), \quad (28)
\end{aligned}
$$

where $B_{k,t} = \alpha_k\left(\Phi^\top D\Phi\right)\prod_{l=k+1}^{t}(I - \alpha_l\Phi^\top D\Phi)$ and $\tilde{B}_{k,t} = \prod_{l=k+1}^{t}\left(I - \alpha_l\Phi^\top D\Phi\right)$.

For simplicity of notation, we define $D' = \Phi^\top D\Phi \in \mathbb{R}^{\mathcal{M}\times\mathcal{M}}$. Notice that $D'$ is a diagonal matrix in $\mathbb{R}^{\mathcal{M}\times\mathcal{M}}$ with the $j$'th entry $d'_j = \sum_{j\in h^{-1}(i)} d_i$. Clearly, $B_{k,t}$ and $\tilde{B}_{k,t}$ are $m$-by-$m$ diagonal matrices, with the $i$'th diagonal entry given by $b_{k,t,i}$ and $\tilde{b}_{k,t,i}$, where $b_{k,t,i} = \alpha_k d'_i \prod_{l=k+1}^{t}(1 - \alpha_l d'_i)$ and $\tilde{b}_{k,t,i} = \prod_{l=k+1}^{t}(1 - \alpha_l d'_i)$. Therefore, for any $i \in \mathcal{M}$, we have

$$
\tilde{b}_{\tau-1,t,i} + \sum_{k=\tau}^{t} b_{k,t,i} = 1. \quad (29)
$$

Also, by the definition of $\sigma'$, we have that for any $i$, almost surely

$$b_{k,t,i} \leq \beta_{k,t} := \alpha_k \prod_{l=k+1}^{t} (1 - \alpha_l \sigma'), \tilde{b}_{k,t,i} \leq \tilde{\beta}_{k,t} = \prod_{l=k+1}^{t} (1 - \alpha_l \sigma'),$$

where $\sigma' = \min\{d_1', \cdots, d_m'\}$.

Recall that $x^*$ is the unique solution of the equation $\Pi F(\Phi x^*) = x^*$. Lemma C.1 shows that we can expand the error term $\|x(t) - x^*\|_v$ recursively.

**Lemma C.1.** *Let* $\Upsilon_t = \|x(t) - x^*\|_v$*, we have almost surely,*

$$\Upsilon_{t+1} \leq \tilde{\beta}_{\tau-1,t} \Upsilon_\tau + \gamma \sup_{i \in \mathcal{M}} \sum_{k=\tau}^{t} b_{k,t,i} \Upsilon_k + \left\| \sum_{k=\tau}^{t} \alpha_k \tilde{B}_{k,t} \epsilon(k) \right\|_v + \left\| \sum_{k=\tau}^{t} \alpha_k \tilde{B}_{k,t} \psi(k) \right\|_v.$$

*Proof of Lemma C.1.* By (28) and the triangle inequality of $\|\cdot\|_v$, we have

$$\|x(t+1) - x^*\|_v$$

$$\leq \sup_{i \in \mathcal{M}} \frac{1}{v_i} \left| \tilde{b}_{\tau-1,t,i} x_i(\tau) + \sum_{k=\tau}^{t} b_{k,t,i} (\Pi F(\Phi x(k)))_i - x_i^* \right|$$

$$+ \left\| \sum_{k=\tau}^{t} \alpha_k \tilde{B}_{k,t} \epsilon(k) \right\|_v + \left\| \sum_{k=\tau}^{t} \alpha_k \tilde{B}_{k,t} \psi(k) \right\|_v. \tag{30}$$

We also see that for each $i \in \mathcal{M}$,

$$\frac{1}{v_i} \left| \tilde{b}_{\tau-1,t,i} x_i(\tau) + \sum_{k=\tau}^{t} b_{k,t,i} (\Pi F(\Phi x(k)))_i - x_i^* \right|$$

$$\leq \tilde{b}_{\tau-1,t,i} \frac{1}{v_i} |x_i(\tau) - x_i^*| + \sum_{k=\tau}^{t} b_{k,t,i} \frac{1}{v_i} |(\Pi F(\Phi x(k)))_i - x_i^*| \tag{31a}$$

$$\leq \tilde{b}_{\tau-1,t,i} \|x(\tau) - x^*\|_v + \sum_{k=\tau}^{t} b_{k,t,i} \|(\Pi F(\Phi x(k))) - x^*\|_v$$

$$\leq \tilde{b}_{\tau-1,t,i} \|x(\tau) - x^*\|_v + \gamma \sum_{k=\tau}^{t} b_{k,t,i} \|x(k) - x^*\|_v, \tag{31b}$$

where in (31a), we use (29) which says $\tilde{b}_{\tau-1,t,i} + \sum_{k=\tau}^{t} b_{k,t,i} = 1$ holds for all $i \in \mathcal{M}$; in (31b), we use Proposition C.1, which says $\Pi F(\Phi \cdot)$ is $\gamma$-contraction in $\|\cdot\|_v$ with fixed point $x^*$.

Therefore, by substituting (31) into (30), we obtain that

$$\Upsilon_{t+1} \leq \tilde{\beta}_{\tau-1,t} \Upsilon_\tau + \gamma \sup_{i \in \mathcal{M}} \sum_{k=\tau}^{t} b_{k,t,i} \Upsilon_k + \left\| \sum_{k=\tau}^{t} \alpha_k \tilde{B}_{k,t} \epsilon(k) \right\|_v + \left\| \sum_{k=\tau}^{t} \alpha_k \tilde{B}_{k,t} \psi(k) \right\|_v.$$

$\square$

**Step 2: Bounding** $\left\| \sum_{k=\tau}^{t} \alpha_k \tilde{B}_{k,t} \epsilon(k) \right\|_v$ **and** $\left\| \sum_{k=\tau}^{t} \alpha_k \tilde{B}_{k,t} \psi(k) \right\|_v$.

We start with a bound on each individual $\epsilon_1(k), \epsilon_2(k)$, and $\psi(k)$ in Lemma C.2. For simplicity of notation, we define $\underline{v} := \inf_{j \in \mathcal{M}} v_j$.

**Lemma C.2.** *The following bounds hold almost surely.*

1. $\|\epsilon_1(t)\|_v \leq 4\bar{x} + 2C + \frac{\bar{w}}{\underline{v}} := \bar{\epsilon}.$

2. $\|\epsilon_2(t)\|_v \leq (2\bar{x} + C) \cdot 2K_1 \exp(-\tau/K_2).$

3. $\|\psi(t)\|_v \leq 3\left(2\bar{x} + C + \frac{\bar{w}}{\underline{v}}\right) \sum_{k=t-\tau+1}^{t} \alpha_{k-1}.$

*Proof of Lemma C.2.* By the definition of $\|\cdot\|_v$ in $\mathbb{R}^{\mathcal{M}}$ and its extension to $\mathbb{R}^{\mathcal{N}}$, the induced matrix norm of $\|\cdot\|$ for a matrix $A = [a_{ij}]_{i \in \mathcal{M}, j \in \mathcal{N}}$ is given by $\|A\|_v = \sup_{i \in \mathcal{M}} \sum_{j \in \mathcal{N}} \frac{v_{h(j)}}{v_i} |a_{ij}|$. Recall that the $i$'th entry of the diagonal matrix $D_t$ is given by $d_{t,i} = \mathbb{P}(i_t = i \mid \mathcal{F}_{t-\tau})$. Hence we have that

$$\left\| \Phi^\top (e_{i_t} e_{i_t}^\top - D_t) \right\|_v = \sup_{j \in \mathcal{M}} \sum_{i \in \mathcal{N}} 1(h(i) = j) \cdot |1(i = i_t) - d_{t,i}| \leq 2. \tag{32}$$

Therefore, we can upper bound $\|\epsilon_1(t)\|_v$ by

$$\|\epsilon_1(t)\|_v = \left\| \Phi^\top \left[ (e_{i_t} e_{i_t}^\top - D_t)(F(\Phi x(t - \tau)) - \Phi x(t - \tau)) + w(t) e_{i_t} \right] \right\|_v$$

$$\leq \left\| \Phi^\top (e_{i_t} e_{i_t}^\top - D_t) \right\|_v \|F(\Phi x(t - \tau)) - \Phi x(t - \tau)\|_v + |w(t)| \left\| \Phi^\top e_{i_t} \right\|_v$$

$$\leq 2 \|F(\Phi x(t - \tau)) - \Phi x(t - \tau)\|_v + |w(t)| \left\| \Phi^\top e_{i_t} \right\|_v \tag{33a}$$

$$\leq 2 \|F(\Phi x(t - \tau))\|_v + 2 \|x(t - \tau)\|_v + \frac{\bar{w}}{\underline{v}} \tag{33b}$$

$$\leq 4\bar{x} + 2C + \frac{\bar{w}}{\underline{v}}, \tag{33c}$$

where we use (32) in (33a); the triangle inequality, the definition of $\bar{v}$, and Assumption 3.3 in (33b); Assumption 3.2 in (33c).

For $\|\epsilon_2(t)\|_v$, recall that

$$\|\epsilon_2(t)\|_v = \left\| \Phi^\top (D_t - D)(F(\Phi x(t - \tau)) - \Phi x(t - \tau)) \right\|_v$$

$$= \sup_{j \in \mathcal{M}} \frac{1}{v_j} \left| \sum_{i \in \mathcal{N}} 1(h(i) = j)(d_{t,i} - d_i)(F(\Phi x(t - \tau)) - \Phi x(t - \tau))_i \right|$$

$$= \sup_{j \in \mathcal{M}} \frac{1}{v_j} \left| \sum_{i \in h^{-1}(j)} (d_{t,i} - d_i)(F(\Phi x(t - \tau)) - \Phi x(t - \tau))_i \right|. \tag{34}$$

By Assumption 3.1, we have that

$$\sup_{\mathcal{S} \subseteq \mathcal{N}} \left| \sum_{i \in \mathcal{S}} d_i - \sum_{i \in \mathcal{S}} d_{t,i} \right| \leq K_1 \exp(-\tau / K_2). \tag{35}$$

Our objective is to bound the following term in (34) for all $j \in \mathcal{M}$:

$$\left| \sum_{i \in h^{-1}(j)} (d_{t,i} - d_i)(F(\Phi x(t - \tau)) - \Phi x(t - \tau))_i \right|.$$

Let $M_j := \sup_{i \in h^{-1}(j)} |(F(\Phi x(t - \tau)) - \Phi x(t - \tau))_i|$. Define function $g : [-M_j, M_j]^{\mathcal{N}} \to \mathbb{R}$ as

$$g(y) = \left| \sum_{i \in h^{-1}(j)} (d_{t,i} - d_i) y_i \right|.$$

Suppose $y_{max} \in \arg\max_y g(y)$. We know that for $i \in h^{-1}(j)$, $(y_{max})_i$ is either $M_j$ or $-M_j$ if $d_{t,i} - d_i \neq 0$. Let $S_j := \{i \in h^{-1}(j) \mid (y_{max})_i = M_j\}$ and $S'_j := \{i \in h^{-1}(j) \mid (y_{max})_i = -M_j\}$. Therefore, we see that

$$\left| \sum_{i \in h^{-1}(j)} (d_{t,i} - d_i)(F(\Phi x(t - \tau)) - \Phi x(t - \tau))_i \right|$$

$$\leq \max_{y \in [-M_j, M_j]^{\mathcal{N}}} g(y) \tag{36a}$$

$$= \left| \sum_{i \in S_j} (d_{t,i} - d_i) \right| M_j + \left| \sum_{i \in S'_j} (d_{t,i} - d_i) \right| M_j$$

$$\leq 2 K_1 \exp(-\tau / K_2) M_j. \tag{36b}$$

where we use the definition of function $g$ in (36a); we use (35) in (36b).

Substituting (36) into (34) gives that

$$
\begin{aligned}
\|\epsilon_2(t)\|_v &\leq \|F(\Phi x(t-\tau)) - \Phi x(t-\tau)\|_v \cdot 2K_1 \exp(-\tau/K_2) \\
&\leq (\|F(\Phi x(t-\tau))\|_v + \|\Phi x(t-\tau)\|_v) \cdot 2K_1 \exp(-\tau/K_2) \quad\quad (37a) \\
&\leq (2\bar{x} + C) \cdot 2K_1 \exp(-\tau/K_2), \quad\quad (37b)
\end{aligned}
$$

where we use the triangle inequality in (37a); we use Assumption 3.2 in (37b).

As for $\|\psi(t)\|_v$, we have the following bound

$$
\begin{aligned}
&\|\psi(t)\|_v \\
&= \left\| \Phi^\top(e_{i_t} e_{i_t}^\top - D)(F(\Phi x(t)) - F(\Phi x(t-\tau))) - \Phi^\top(e_{i_t} e_{i_t}^\top - D)\Phi(x(t) - x(t-\tau)) \right\|_v \\
&\leq \left\| \Phi^\top(e_{i_t} e_{i_t}^\top - D)(F(\Phi x(t)) - F(\Phi x(t-\tau))) \right\|_v + \left\| \Phi^\top(e_{i_t} e_{i_t}^\top - D)\Phi(x(t) - x(t-\tau)) \right\|_v \\
&\leq \left\| \Phi^\top(e_{i_t} e_{i_t}^\top - D) \right\|_v \cdot \|(F(\Phi x(t)) - F(\Phi x(t-\tau)))\|_v \\
&\quad + \left\| \Phi^\top(e_{i_t} e_{i_t}^\top - D)\Phi \right\|_v \cdot \|(x(t) - x(t-\tau))\|_v. \quad\quad (38)
\end{aligned}
$$

Notice that

$$
\left\| \Phi^\top(e_{i_t} e_{i_t}^\top - D)\Phi \right\|_v = \left\| \xi_{h(i_t)} \xi_{h(i_t)}^\top - D' \right\|_v = \sup_{j \in \mathcal{M}} \left| \mathbb{1}(h(i_t) = j) - d_j' \right| \leq 1.
$$

Substituting this into (38) and use (32), we obtain that

$$
\begin{aligned}
\|\psi(t)\|_v &\leq 2\|F(\Phi x(t)) - F(\Phi x(t-\tau))\|_v + \|x(t) - x(t-\tau)\|_v \\
&\leq 3\|x(t) - x(t-\tau)\|_v \\
&\leq 3 \sum_{k=t-\tau+1}^{t} \|x(k) - x(k-1)\|_v. \quad\quad (39)
\end{aligned}
$$

By the update rule of $x$ and Assumption 3.2, we have that

$$
\begin{aligned}
\|x(t) - x(t-1)\|_v &\leq \alpha_{t-1} \left( \|F(\Phi x(t-1))\|_v + \|x(t-1)\|_v + \frac{\bar{w}}{\underline{v}} \right) \\
&\leq \alpha_{t-1} \left( 2\bar{x} + C + \frac{\bar{w}}{\underline{v}} \right). \quad\quad (40)
\end{aligned}
$$

Substituting (40) into (39), we obtain that

$$
\|\psi(t)\|_v \leq 3 \left( 2\bar{x} + C + \frac{\bar{w}}{\underline{v}} \right) \sum_{k=t-\tau+1}^{t} \alpha_{k-1}.
$$

$\square$

**Lemma C.3.** *If $\alpha_t = \frac{H}{t+t_0}$, where $H > \frac{2}{\sigma'}$ and $t_0 \geq \max(4H, \tau)$, then $\beta_{k,t}, \tilde{\beta}_{k,t}$ satisfies the following*

1. $\beta_{k,t} \leq \frac{H}{k+t_0} \left( \frac{k+1+t_0}{t+1+t_0} \right)^{\sigma' H}, \tilde{\beta}_{k,t} \leq \left( \frac{k+1+t_0}{t+1+t_0} \right)^{\sigma' H}.$

2. $\sum_{k=1}^{t} \beta_{k,t}^2 \leq \frac{2H}{\sigma'} \frac{1}{t+1+t_0}.$

3. $\sum_{k=\tau}^{t} \beta_{k,t} \sum_{l=k-\tau+1}^{k} \alpha_{l-1} \leq \frac{8H\tau}{\sigma'} \frac{1}{t+1+t_0}.$

*Proof of Lemma C.3.* To show Lemma C.3, we only need to substitute $\sigma'$ for $\sigma$ in the proof of [37][Lemma 10]. $\square$

**Lemma C.4.** *The following inequality holds almost surely*

$$
\left\| \sum_{k=\tau}^{t} \alpha_k \tilde{B}_{k,t} \psi(k) \right\|_v \leq \frac{24 \left( 2\bar{x} + C + \frac{\bar{w}}{\underline{v}} \right) H\tau}{\sigma'} \frac{1}{t+1+t_0} := C_\psi \frac{1}{t+1+t_0}.
$$

*Proof of Lemma C.4.* We have that

$$\left\| \sum_{k=\tau}^{t} \alpha_k \tilde{B}_{k,t} \psi(k) \right\|_v \leq \sum_{k=\tau}^{t} \alpha_k \left\| \tilde{B}_{k,t} \right\|_v \|\psi(k)\|_v$$

$$\leq 3\left( 2\bar{x} + C + \frac{\bar{w}}{\underline{v}} \right) \sum_{k=\tau}^{t} \beta_{k,t} \sum_{l=k-\tau+1}^{k} \alpha_{l-1} \tag{41a}$$

$$\leq \frac{24\left( 2\bar{x} + C + \frac{\bar{w}}{\underline{v}} \right) H\tau}{\sigma'} \frac{1}{t+1+t_0}, \tag{41b}$$

where we use Lemma C.2 in (41a); Lemma C.3 in (41b). $\qquad\square$

**Lemma C.5.** *For each $t$, with probability at least $1 - \delta$, we have*

$$\left\| \sum_{k=\tau}^{t} \alpha_k \tilde{B}_{k,t} \epsilon_1(k) \right\|_v \leq \frac{H\bar{\epsilon}}{t+t_0} \sqrt{2\tau t \log\left( \frac{2\tau m}{\delta} \right)}.$$

To show Lemma C.5, we need to use Lemma C.6, which is Lemma 13 in [37].

**Lemma C.6.** *Let $X_t$ be a $\mathcal{F}_t$-adapted stochastic process which satisfies $\mathbb{E}X_t \mid \mathcal{F}_{t-\tau} = 0$. Further, $|X_t| \leq \bar{X}_t$ almost surely. Then with probability $1 - \delta$, we have, $\left| \sum_{k=0}^{t} X_t \right| \leq \sqrt{2\tau \sum_{k=0}^{t} \bar{X}_k^2 \log\left( \frac{2\tau}{\delta} \right)}$.*

*Proof of Lemma C.5.* Recall that $\sum_{k=\tau}^{t} \alpha_k \tilde{B}_{k,t} \epsilon_1(k)$ is a random vector in $\mathbb{R}^{\mathcal{M}}$, with its $i$'th entry

$$\sum_{k=\tau}^{t} \alpha_k (\epsilon_1)_i(k) \prod_{l=k+1}^{t} (1 - \alpha_l d_i').$$

Since step sizes $\{\alpha_l\}$ are deterministic, we see that

$$\mathbb{E}\left[ \alpha_k (\epsilon_1)_i(k) \prod_{l=k+1}^{t} (1 - \alpha_l d_i') \mid \mathcal{F}_{k-\tau} \right] = \alpha_k \prod_{l=k+1}^{t} (1 - \alpha_l d_i') \mathbb{E}[(\epsilon_1)_i(k) \mid \mathcal{F}_{k-\tau}] = 0.$$

Notice that

$$\alpha_k \prod_{l=k+1}^{t} (1 - \alpha_l d_i') = \frac{H}{k+t_0} \prod_{l=k+1}^{t} \left( 1 - \frac{H d_i'}{l+t_0} \right) \tag{42a}$$

$$\leq \frac{H}{k+t_0} \prod_{l=k+1}^{t} \left( 1 - \frac{2}{l+t_0} \right) \tag{42b}$$

$$\leq \frac{H}{k+t_0} \prod_{l=k+1}^{t} \left( 1 - \frac{1}{l+t_0} \right)$$

$$\leq \frac{H}{t+t_0},$$

where we use $\alpha_l = \frac{H}{l+t_0}$ in (42a); we use $H > \frac{2}{\sigma'}$ in (42b).

By the definition of $\bar{\epsilon}$, we also see that $|(\epsilon_1)_i(k)| \leq v_i \bar{\epsilon}$. Therefore, by Lemma C.6, we obtain that

$$\left| \sum_{k=\tau}^{t} \alpha_k (\epsilon_1)_i(k) \prod_{l=k+1}^{t} (1 - \alpha_l d_i') \right| \leq \frac{H v_i \bar{\epsilon}}{t+t_0} \sqrt{2\tau t \log\left( \frac{2\tau}{\delta} \right)}$$

holds with probability at least $1 - \delta$. By union bound, we see that with probability at least $1 - \delta$,

$$\left\| \sum_{k=\tau}^{t} \alpha_k \tilde{B}_{k,t} \epsilon_1(k) \right\|_v \leq \frac{H\bar{\epsilon}}{t+t_0} \sqrt{2\tau t \log\left( \frac{2\tau m}{\delta} \right)}.$$

$\qquad\square$

**Lemma C.7.** *If we set $\tau$ to be an integer such that*

$$\tau \geq 2K_2 \max(\log t, 1),$$

*we have that*

$$\left\| \sum_{k=\tau}^{t} \alpha_k \tilde{B}_{k,t} \epsilon_2(k) \right\|_v \leq \frac{C_{\epsilon_2}}{t + t_0 + 1},$$

*where $t_0 = \max(\tau, 4H)$ and $C_{\epsilon_2} = (2\bar{x} + C) \cdot 2K_1(1 + 2K_2 + 4H)$.*

*Proof of Lemma C.7.* Since $K_2 \geq 1$, the bound is trivial when $t = 1$. We consider the case when $t \geq 2$ below.

Since $\alpha_k \tilde{B}_{k,t}$ is a diagonal matrix and its entries are positive and less than 1, we have that

$$\left\| \sum_{k=\tau}^{t} \alpha_k \tilde{B}_{k,t} \epsilon_2(k) \right\|_v \leq \sum_{k=\tau}^{t} \left\| \alpha_k \tilde{B}_{k,t} \right\|_v \cdot \| \epsilon_2(k) \|_v$$

$$\leq t \| \epsilon_2(k) \|_v \tag{43a}$$

$$\leq t(2\bar{x} + C) \cdot 2K_1 \exp(-\tau/K_2). \tag{43b}$$

where we use $\left\| \alpha_k \tilde{B}_{k,t} \right\|_v \leq 1$ in (43a); Lemma C.2 in (43b).

To show Lemma C.7, we only need to show

$$t(2\bar{x} + C) \cdot 2K_1(t + \tau + 4H) \exp(-\tau/K_2) \leq C_{\epsilon_2} \tag{44}$$

holds for all $\tau \geq 2K_2 \log t$ because $t + t_0 + 1 \leq t + \tau + 4H$.

To study how the left hand side of (44) changes with $\tau$, we define function

$$g(\tau) = (\tau + t + 4H) \exp(-\tau/K_2).$$

Notice that we view $\tau$ as real number in function $g$, so we can get the derivative of $g$:

$$g'(\tau) = \frac{\exp(-\tau/K_2)}{K_2}(K_2 - t - 4H - \tau).$$

Therefore, when $\tau \geq 2K_2 \log t$, we always have $g'(\tau) < 0$. Hence we obtain that

$$g(\tau) \leq g(2K_2 \log t) = \frac{2K_2 \log t + t + 4H}{t^2} \leq \frac{1 + 2K_2 + 4H}{t} \tag{45}$$

holds for all $\tau \geq 2K_2 \log t$.

Substituting (45) into (44) finishes the proof. $\qquad\square$

**Step 3: Bounding the error sequence.** Based on the recursive relationship we derived in Lemma C.1 and the bounds we obtained in Step 2, we want to show that, with probability $1 - \delta$,

$$\Upsilon_t \leq \frac{C_a}{\sqrt{t + t_0}} + \frac{C_a'}{t + t_0}, \tag{46}$$

holds for all $\tau \leq t \leq T$, where

$$C_a = \frac{2H\bar{\epsilon}}{1 - \gamma} \sqrt{2\tau \log\left(\frac{2\tau mT}{\delta}\right)}, C_a' = \frac{4}{1 - \gamma} \max(C_\psi + C_{\epsilon_2}, 2\bar{x}(\tau + t_0)).$$

Notice that $C_a$ and $C_a'$ are independent of $t$ but may dependent on $T$. We set $\tau = 2K_2 \log T$.

By applying union bound to Lemma C.5, we see that with probability at least $1 - \delta$, for any $t \leq T$,

$$\left\| \sum_{k=\tau}^{t} \alpha_k \tilde{B}_{k,t} \epsilon_1(k) \right\|_v \leq \frac{C_{\epsilon_1}}{\sqrt{t + 1 + t_0}},$$

where $C_{\epsilon_1} = H\bar{\epsilon}\sqrt{2\tau \log\left(\frac{2\tau mT}{\delta}\right)}$.

Therefore, we get with probability $1 - \delta$, (47) holds for all $\tau \leq t \leq T$:

$$\Upsilon_{t+1} \leq \tilde{\beta}_{\tau-1,t}\Upsilon_\tau + \gamma \sup_{i \in \mathcal{M}} \sum_{k=\tau}^{t} b_{k,t,i}\Upsilon_k + \frac{C_{\epsilon_1}}{\sqrt{t+1+t_0}} + \frac{C_\psi + C_{\epsilon_2}}{t+1+t_0}. \tag{47}$$

We now condition on (47) to show (46) by induction. (46) is true for $t = \tau$, as $\frac{C'_a}{\tau+t_0} \geq \frac{8}{1-\gamma}\bar{x} \geq \Upsilon_\tau$, where we have used $\Upsilon_\tau = \|x(\tau) - x^*\|_v \leq \|x(\tau)\|_v + \|x^*\|_v \leq 2\bar{x}$. Then, assuming (46) is true for up to $k \leq t$. By (47), we have that

$$\begin{aligned}
\Upsilon_{t+1} &\leq \tilde{\beta}_{\tau-1,t}\Upsilon_\tau + \gamma \sup_{i \in \mathcal{M}} \sum_{k=\tau}^{t} b_{k,t,i}\left[\frac{C_a}{\sqrt{k+t_0}} + \frac{C'_a}{k+t_0}\right] + \frac{C_{\epsilon_1}}{\sqrt{t+1+t_0}} + \frac{C_\psi + C_{\epsilon_2}}{t+1+t_0} \\
&\leq \tilde{\beta}_{\tau-1,t}\Upsilon_\tau + \gamma C_a \sup_{i \in \mathcal{M}} \sum_{k=\tau}^{t} b_{k,t,i}\frac{1}{\sqrt{k+t_0}} + \gamma C'_a \sup_{i \in \mathcal{M}} \sum_{k=\tau}^{t} \frac{1}{k+t_0}b_{k,t,i} \\
&\quad + \frac{C_{\epsilon_1}}{\sqrt{t+1+t_0}} + \frac{C_\psi + C_{\epsilon_2}}{t+1+t_0}. \tag{48}
\end{aligned}$$

We use the following auxiliary lemma to handle the second and the third term in (48).

**Lemma C.8.** *If $\sigma'H(1 - \sqrt{\gamma}) \geq 1, t_0 \geq 1$, and $\alpha_0 \leq \frac{1}{2}$, then, for any $i \in \mathcal{N}$, and any $0 < \omega \leq 1$, we have*

$$\sum_{k=\tau}^{t} b_{k,t,i}\frac{1}{(k+t_0)^\omega} \leq \frac{1}{\sqrt{\gamma}(t+1+t_0)^\omega}.$$

*Proof of Lemma C.8.* Recall that $\alpha_k = \frac{H}{k+t_0}$, and $b_{k,t,i} = \alpha_k d'_i \prod_{l=k+1}^{t}(1 - \alpha_l d'_i)$, where $d'_i \geq \sigma'$.

Define $e_t = \sum_{k=\tau}^{t} b_{k,t,i}\frac{1}{(k+t_0)^\omega}$. We use induction on $t$ to show that $e_t \leq \frac{1}{\sqrt{\gamma}(t+1+t_0)^\omega}$.

The statement is clearly true for $t = \tau$. Assume it is true for $t - 1$. Notice that

$$\begin{aligned}
e_t &= \sum_{k=\tau}^{t-1} b_{k,t,i}\frac{1}{(k+t_0)^\omega} + b_{t,t,i}\frac{1}{(t+t_0)^\omega} \\
&= (1 - \alpha_t d'_i)\sum_{k=\tau}^{t-1} b_{k,t-1,i}\frac{1}{(k+t_0)^\omega} + \alpha_t d'_i\frac{1}{(t+t_0)^\omega} \tag{49a} \\
&= (1 - \alpha_t d'_i)e_{t-1} + \alpha_t d'_i\frac{1}{(t+t_0)^\omega} \\
&\leq (1 - \alpha_t d'_i)\frac{1}{\sqrt{\gamma}(t+t_0)^\omega} + \alpha_t d'_i\frac{1}{(t+t_0)^\omega} \tag{49b} \\
&= [1 - \alpha_t d'_i(1 - \sqrt{\gamma})]\frac{1}{\sqrt{\gamma}(t+t_0)^\omega},
\end{aligned}$$

where we use $b_{t,t,i} = \alpha_t d'_i$ in (49a); we use the induction assumption in (49b).

Plugging in $\alpha_t = \frac{H}{t+t_0}$, we see that

$$e_t \leq \left[ 1 - \frac{\sigma' H}{t + t_0}(1 - \sqrt{\gamma}) \right] \frac{1}{\sqrt{\gamma}(t + t_0)^\omega} \tag{50a}$$

$$= \left[ 1 - \frac{\sigma' H}{t + t_0}(1 - \sqrt{\gamma}) \right] \left( 1 + \frac{1}{t + t_0} \right)^\omega \frac{1}{\sqrt{\gamma}(t + 1 + t_0)^\omega}$$

$$\leq \left( 1 - \frac{1}{t + t_0} \right) \left( 1 + \frac{1}{t + t_0} \right)^\omega \frac{1}{\sqrt{\gamma}(t + 1 + t_0)^\omega} \tag{50b}$$

$$\leq \left( 1 - \frac{1}{t + t_0} \right) \left( 1 + \frac{1}{t + t_0} \right) \frac{1}{\sqrt{\gamma}(t + 1 + t_0)^\omega} \tag{50c}$$

$$\leq \frac{1}{\sqrt{\gamma}(t + 1 + t_0)^\omega},$$

where we use $d_i' \geq \sigma'$ in (50a); we use the assumption that $\sigma' H(1 - \sqrt{\gamma}) \geq 1$ in (50b); we use $0 < \omega \leq 1$ in (50c). $\qquad\square$

Applying Lemma C.8 to (48), we see that

$$\Upsilon_{t+1} \leq \tilde{\beta}_{\tau-1,t} \Upsilon_\tau + \sqrt{\gamma} C_a \frac{1}{\sqrt{t + 1 + t_0}} + \sqrt{\gamma} C_a' \frac{1}{t + 1 + t_0}$$

$$+ C_{\epsilon_1} \frac{1}{\sqrt{t + 1 + t_0}} + (C_\psi + C_{\epsilon_2}) \frac{1}{t + 1 + t_0} \tag{51a}$$

$$\leq \left( \sqrt{\gamma} C_a \frac{1}{\sqrt{t + 1 + t_0}} + C_{\epsilon_1} \frac{1}{\sqrt{t + 1 + t_0}} \right)$$

$$+ \left( \sqrt{\gamma} C_a' \frac{1}{t + 1 + t_0} + (C_\psi + C_{\epsilon_2}) \frac{1}{t + 1 + t_0} + \left( \frac{\tau + t_0}{t + 1 + t_0} \right)^{\sigma' H} \Upsilon_\tau \right), \tag{51b}$$

where we use Lemma C.8 in (51a); we use the bound on $\tilde{\beta}_{\tau-1,t}$ in Lemma C.3 in (51b).

To bound the two terms in (51b), we define

$$\chi_t := \sqrt{\gamma} C_a \frac{1}{\sqrt{t + 1 + t_0}} + C_{\epsilon_1} \frac{1}{\sqrt{t + 1 + t_0}}$$

and

$$\chi_t' = \sqrt{\gamma} C_a' \frac{1}{t + 1 + t_0} + (C_\psi + C_{\epsilon_2}) \frac{1}{t + 1 + t_0} + \left( \frac{\tau + t_0}{t + 1 + t_0} \right)^{\sigma' H} a_\tau.$$

To finish the induction, it suffices to show that $\chi_t \leq \frac{C_a}{\sqrt{t+1+t_0}}$ and $\chi_t' \leq \frac{C_a'}{t+1+t_0}$. To see this

$$\chi_t \frac{\sqrt{t + 1 + t_0}}{C_a} = \sqrt{\gamma} + \frac{C_{\epsilon_1}}{C_a}, \chi_t' \frac{t + 1 + t_0}{C_a'} = \sqrt{\gamma} + \frac{C_\psi + C_{\epsilon_2}}{C_a'} + \frac{\Upsilon_\tau(\tau + t_0)}{C_a'} \left( \frac{\tau + t_0}{t + 1 + t_0} \right)^{\sigma' H - 1}.$$

It suffices to show that $\frac{C_{\epsilon_1}}{C_a} \leq 1 - \sqrt{\gamma}$, $\frac{C_\psi + C_{\epsilon_2}}{C_a'} \leq \frac{1 - \sqrt{\gamma}}{2}$, and $\frac{\Upsilon_\tau(\tau+t_0)}{C_a'} \leq \frac{1 - \sqrt{\gamma}}{2}$. Recall that

$$C_a = \frac{2H\bar{\epsilon}}{1 - \gamma} \sqrt{2\tau \log\left( \frac{2\tau m T}{\delta} \right)}, C_a' = \frac{4}{1 - \gamma} \max(C_\psi + C_{\epsilon_2}, 2\bar{x}(\tau + t_0)),$$

and

$$C_{\epsilon_1} = H\bar{\epsilon} \sqrt{2\tau \log\left( \frac{2\tau m T}{\delta} \right)}.$$

Using that $\Upsilon_\tau \leq 2\bar{x}$, one can check that $C_a$ and $C_a'$ satisfy the above three inequalities.

## C.3 Parameter Upper Bound

**Proposition C.2.** *Suppose Assumptions 3.2 and 3.3 hold. Then for all t,*

$$\|x(t)\|_v \leq \frac{1}{1-\gamma}\left((1+\gamma)\|y^*\|_v + \frac{\bar{w}}{\underline{v}}\right)$$

*holds almost surely, where $y^* \in \mathbb{R}^{\mathcal{N}}$ is the stationary point of $F$.*

*Proof of Proposition C.2.* By Assumption 3.2, we have that for all $x \in \mathbb{R}^{\mathcal{M}}$,

$$\|F(\Phi x)\|_v \leq \|F(\Phi x) - F(y^*)\|_v + \|F(y^*)\|_v \tag{52a}$$
$$\leq \gamma\|\Phi x - y^*\|_v + \|y^*\|_v \tag{52b}$$
$$\leq \gamma\|x\|_v + (1+\gamma)\|y^*\|_v, \tag{52c}$$

where we use the triangle inequality in (52a) and (52c); we use Assumption 3.2 in (52b).

Let $\bar{x} = \frac{1}{1-\gamma}\left((1+\gamma)\|y^*\|_v + \frac{\bar{w}}{\underline{v}}\right)$. We prove $\|x(t)\|_v \leq \bar{x}$ by induction on $t$. Since we initialize $x(0)$ to be $\mathbf{0}$, the statement is true for $t = 0$.

Suppose the statement is true for $t$. By the update rule of $x$, we see that

$$\frac{1}{v_{h(i_t)}}\left|x_{h(i_t)}(t+1)\right| \leq (1-\alpha_t)\frac{1}{v_{h(i_t)}}\left|x_{h(i_t)}(t)\right| + \alpha_t\left(\frac{1}{v_{h(i_t)}}|F_{i_t}(\Phi x(t))| + \frac{1}{v_{h(i_t)}}|w(t)|\right)$$

$$\leq (1-\alpha_t)\|x(t)\|_v + \alpha_t\left(\|F(\Phi x(t))\|_v + \frac{\bar{w}}{\underline{v}}\right) \tag{53a}$$

$$\leq (1-\alpha_t)\|x(t)\|_v + \alpha_t\left(\gamma\|x(t)\|_v + (1+\gamma)\|y^*\|_v + \frac{\bar{w}}{\underline{v}}\right) \tag{53b}$$

$$\leq (1-\alpha_t)\bar{x} + \alpha_t\left(\gamma\bar{x} + (1+\gamma)\|y^*\|_v + \frac{\bar{w}}{\underline{v}}\right) \tag{53c}$$

$$= \bar{x},$$

where we use Assumption 3.3 in (53a); (52) in (53b); the induction assumption in (53c).

For $j \neq h(i_t), j \in \mathcal{M}$, we have that

$$\frac{1}{v_j}|x_j(t+1)| = \frac{1}{v_j}|x_j(t)| \leq \|x(t)\|_v \leq \bar{x}. \tag{54}$$

Combining (53) and (54), we see that the statement also holds for $t+1$. Hence we have showed $\|x(t)\|_v \leq \bar{x}$ by induction. $\qquad\square$

# D  TD/Q-Learning with State Aggregation

## D.1  Asymptotic Convergence of TD Learning with State Aggregation

Our asymptotic convergence result for TD learning with state aggregation builds upon the asymptotic convergence result for TD learning with linear function approximation shown in [49]. For completeness, we first present the main result of [49] in Theorem D.1. In order to do this, we must first state a few definitions and assumptions made in [49].

We use $\phi(i) \in \mathbb{R}^m$ to denote the feature vector associated with state $i \in \mathcal{N}$. Feature matrix $\Phi$ is a $n$-by-$m$ matrix whose $i$'th row is $\phi(i)^\top$. Starting from $\theta(0) = \mathbf{0}$, the $TD(\lambda)$ algorithm keeps updating $\theta, \psi$ by the following update rule,

$$\theta(t+1) = \theta(t) + \alpha_t d_t \psi_t,$$
$$\psi_{t+1} = \gamma\lambda\psi_t + \phi(i_{t+1}),$$

where $\psi_t$ is named *eligible vector* in [49] and satisfies $\psi_0 = \phi(i_0)$.

Recall that $D = diag(d_1, d_2, \cdots, d_n)$ denotes the stationary distribution of Markov chain $\{i_t\}$. For vectors $x, y \in \mathbb{R}^n$, we define inner product $\langle x, y \rangle = x^\top D y$. The induced norm of this inner product is $\|\cdot\|_D = \sqrt{\langle \cdot, \cdot \rangle_D}$. Let $L_2(\mathcal{N}, D)$ denote the set of vectors $V \in \mathbb{R}^n$ such that $\|V\|_D$ is finite.

Recall that we define $\Pi = (\Phi^\top D \Phi)^{-1} \Phi^\top D$. As shown in [49], the projection matrix that projects an arbitrary vector in $\mathbb{R}^n$ to the set $\{\Phi\theta \mid \theta \in \mathbb{R}^m\}$ is given by $\Phi\Pi$, i.e. for any $V \in L_2(\mathcal{N}, D)$, we have

$$\Phi\Pi V = \underset{\bar{V} \in \{\Phi\theta|\theta \in \mathbb{R}^m\}}{\arg\min} \left\|V - \bar{V}\right\|_D.$$

Notice that our definition of matrix $\Pi$ is slightly different with [49] because we want to be consistent with Section 3.1.

To characterize the TD($\lambda$) algorithm's dynamics, [49] defines $T^{(\lambda)} : L_2(\mathcal{N}, D) \rightarrow L_2(\mathcal{N}, D)$ operator as following: for all $V \in \mathbb{R}^n$, let the $i$'th dimension of $\left(T^{(\lambda)}V\right)$ be defined as

$$\left(T^{(\lambda)}V\right)_i = \begin{cases} (1-\lambda)\sum_{m=0}^{\infty}\lambda^m\mathbb{E}\left[\sum_{t=0}^{m}\gamma^t r(i_t, i_{t+1}) + \gamma^{m+1}V_{i_{m+1}} \mid i_0 = i\right] & \text{if } \lambda < 1 \\ \mathbb{E}[\sum_{t=0}^{\infty}\gamma^t r(i_t, i_{t+1}) \mid i_0 = i] & \text{if } \lambda = 1. \end{cases}$$

If $V$ is an approximation of the value function $V^*$, $T^{(\lambda)}$ can be viewed as an improved approximation to $V^*$. Notice that when $\lambda = 0$, $T^{(\lambda)}$ is identical with the Bellman operator.

Formally, [49] made four necessary assumptions for their main result (Theorem D.1). We omit the third assumption ([49][Assumption 3]) in our summary because it must hold when the state space $\mathcal{N}$ is finite.

The first assumption ([49][Assumption 1]) concerns the stationary distribution and the reward function of the Markov chain $\{i_t\}$. It must hold when Assumption 3.1 holds and every stage reward $r_t$ is upper bounded by $\bar{r}$, as assumed by Theorem 3.2.

**Assumption D.1.** *The transition probability and cost function satisfies the following two conditions:*

1. *The Markov chain $\{i_t\}$ is irreducible and aperiodic. Furthermore, there is a unique distribution $d$ that satisfies $d^\top P = d^\top$ with $d_i > 0$ for all $i \in \mathcal{N}$. Let $\mathbb{E}_0$ stand for expectation with respect to this distribution.*

2. *The reward function $r(i_t, i_{t+1})$ satisfies $\mathbb{E}_0\left[r^2(i_t, i_{t+1})\right] < \infty$.*

The second assumption ([49][Assumption 2]) concerns the feature vectors and the feature matrix. It must hold when $\Phi$ is defined as (4).

**Assumption D.2.** *The following two conditions hold for $\Phi$:*

1. *The matrix $\Phi$ has full column rank; that is, the $m$ columns (named basis functions in [49]) $\{\phi_k \mid k = 1, \cdots, m\}$ are linearly independent.*

2. *For every $k$, the basis function $\phi_k$ satisfies $\mathbb{E}_0\left[\phi_k^2(i_t)\right] < \infty$.*

The third assumption ([49][Assumption 4]) concerns the learning step size. It must hold if the learning step sizes are as defined in Theorem 3.2.

**Assumption D.3.** *The step sizes $\alpha_t$ are positive, nonincreasing, and chosen prior to execution of the algorithm. Furthermore, they satisfy $\sum_{t=0}^{\infty} \alpha_t = \infty$ and $\sum_{t=0}^{\infty} \alpha_t^2 < \infty$.*

Now we are ready to present the main asymptotic convergence result given in [49].

**Theorem D.1.** *Under Assumptions D.1, D.2, D.3, the following hold.*

1. *The value function $V$ is in $L_2(\mathcal{N}, D)$.*

2. *For any $\lambda \in [0, 1]$, the TD($\lambda$) algorithm with linear function approximation converges with probability one.*

3. *The limit of convergence $\theta^*$ is the unique solution of the equation*

$$\Pi T^{(\lambda)}(\Phi\theta^*) = \theta^*.$$

*4. Furthermore, $\theta^*$ satisfies*

$$\|\Phi\theta^* - V^*\|_D \le \frac{1 - \lambda\gamma}{1 - \gamma}\|\Phi\Pi V^* - V^*\|_D. \tag{55}$$

Notice that (55) is not exactly the result we want to obtain. Specifically, we want the both sides of (55) to be in $\|\cdot\|_\infty$ instead of $\|\cdot\|_D$. Although this kind of result is not obtainable for general TD learning with linear function approximation, we can leverage the special assumptions for state aggregation, which are summarized below:

**Assumption D.4.** *$h : \mathcal{N} \to \mathcal{M}$ is a surjective function from set $\mathcal{N}$ to $\mathcal{M}$. The feature matrix $\Phi$ is as defined in (4), i.e. the feature vector associated with state $i \in \mathcal{N}$ is given by*

$$\phi_k(i) = \begin{cases} 1 & \text{if } k = h(i) \\ 0 & \text{otherwise} \end{cases}, \forall k \in \mathcal{M}.$$

*Further, if $h(i) = h(i')$ for $i, i' \in \mathcal{N}$, we have $|V^*(i) - V^*(i')| \le \zeta$ for a fixed positive constant $\zeta$.*

Under Assumption D.4, we can show the asymptotic error bound in the infinity norm as we desired:

**Theorem D.2.** *Under Assumptions D.1, D.2, D.3, if Assumption D.4 also holds, the limit of convergence $\theta^*$ of the $TD(\lambda)$ algorithm satisfies*

$$\|\Phi\theta^* - V^*\|_\infty \le \frac{(1 - \lambda\gamma)}{1 - \gamma}\|\Phi\Pi V^* - V^*\|_\infty \le \frac{(1 - \lambda\gamma)}{1 - \gamma}\zeta.$$

To show Theorem D.2, we need to prove several auxiliary lemmas first.

**Lemma D.3.** *Under Assumption D.1, for any $V \in L_2(\mathcal{N}, D)$, we have $\|PV\|_\infty \le \|V\|_\infty$.*

*Proof of Lemma D.3.* This lemma holds because the transition matrix $P$ is non-expansive in infinity norm. $\square$

**Lemma D.4.** *Under Assumption D.1, for any $V, \bar{V} \in L_2(\mathcal{N}, D)$, we have*

$$\left\|T^{(\lambda)}V - T^{(\lambda)}\bar{V}\right\|_\infty \le \frac{\gamma(1 - \lambda)}{1 - \gamma\lambda}\|V - \bar{V}\|_\infty.$$

*Proof of Lemma D.4.* By the definition of $T^{(\lambda)}$, we have that

$$\left\|T^{(\lambda)}V - T^{(\lambda)}\bar{V}\right\|_\infty = \left\|(1 - \lambda)\sum_{m=0}^\infty \lambda^m(\gamma P)^{m+1}(V - \bar{V})\right\|_\infty$$

$$\le (1 - \lambda)\sum_{m=0}^\infty \lambda^m \gamma^{m+1}\|V - \bar{V}\|_\infty \tag{56a}$$

$$\frac{\gamma(1 - \lambda)}{1 - \gamma\lambda}\|V - \bar{V}\|_\infty,$$

where inequality (56a) holds because $\|V - \bar{V}\|_\infty < \infty$ so we use Lemma D.3. $\square$

**Lemma D.5.** *Under Assumption D.1 and D.4, we have*

$$\|\Phi\Pi V^* - V^*\|_\infty \le \zeta \tag{57}$$

*and for any $V \in L_2(\mathcal{N}, D)$*

$$\|\Phi\Pi V\|_\infty \le \|V\|_\infty. \tag{58}$$

*Proof of Lemma D.5.* For $j \in \mathcal{M}$, we use $h^{-1}(j) \subseteq \mathcal{N}$ to denote all the elements in $\mathcal{N}$ whose feature is $e_j$, i.e. $h^{-1}(j) = \{i \mid i \in \mathcal{N}, h(i) = j\}$. Since $h$ is surjection, $h^{-1}(j) \ne \emptyset, \forall j \in \mathcal{M}$. Since $\Phi\Pi$ is the projection matrix that projects a vector in $\mathbb{R}^n$ to the set $\{\Phi\theta \mid \theta \in \mathbb{R}^m\}$, we have

$$\Pi V = \arg\min_{\theta \in \mathbb{R}^m} \sum_{j \in \mathcal{M}} \sum_{i \in h^{-1}(j)} d_i(V_i - \theta_j).$$

Hence the optimal $\theta_j$ must be in the range $\left[\min_{i \in h^{-1}(j)} V_i, \max_{i \in h^{-1}(j)} V_i\right]$. Therefore, we see that

$$\left|(\Phi\Pi V)_i\right| = \left|(\Pi V)_{h(i)}\right| \leq \max_{i' \in h^{-1}(h(i))} |V_{i'}|,$$

which shows (58). Besides, we also have

$$\left|(\Phi\Pi V)_i - V_i\right| \leq \max\left(\left|\min_{i' \in h^{-1}(h(i))} V_{i'} - V_i\right|, \left|\max_{i' \in h^{-1}(h(i))} V_{i'} - V_i\right|\right). \tag{59}$$

holds for all $z \in \mathcal{Z}$. Let $V = V^*$ and use Assumption D.4 in (59) gives (57). $\qquad\square$

Now we come back to the proof of Theorem D.2.

Notice that

$$\|\Phi\theta^* - V^*\|_\infty \leq \|\Phi\theta^* - \Phi\Pi V^*\|_\infty + \|\Phi\Pi V^* - V^*\|_\infty \tag{60a}$$

$$= \left\|\Phi\Pi T^{(\lambda)}(\Phi\theta^*) - \Phi\Pi V^*\right\|_\infty + \|\Phi\Pi V^* - V^*\|_\infty \tag{60b}$$

$$\leq \left\|T^{(\lambda)}(\Phi\theta^*) - V^*\right\|_\infty + \|\Phi\Pi V^* - V^*\|_\infty \tag{60c}$$

$$\leq \frac{\gamma(1-\lambda)}{1-\gamma\lambda}\|\Phi\theta^* - V^*\|_\infty + \|\Phi\Pi V^* - V^*\|_\infty, \tag{60d}$$

where we use the triangle inequality in (60a); Theorem D.1 in (60b); Lemma D.5 in (60c); Lemma D.4 in (60d).

Therefore, we obtain that

$$\|\Phi\theta^* - V^*\|_\infty \leq \frac{(1-\lambda\gamma)}{1-\gamma}\|\Pi V^* - V^*\|_\infty \leq \frac{(1-\lambda\gamma)}{1-\gamma}\zeta,$$

where we use Lemma D.5 in the second inequality.

### D.2 Proof of Theorem 3.2

Before presenting the proof of Theorem 3.2, we first show two upper bounds that are needed in the assumptions of Theorem 3.1. We defer the proof of this result to Appendix D.3.

**Proposition D.1.** *Under the same assumptions as Theorem 3.2, we have $\|\theta(t)\|_\infty \leq \bar{\theta} := \frac{\bar{r}}{1-\gamma}$ holds for all $t$ almost surely and $\|\theta^*\|_\infty \leq \bar{\theta}$. $|w(t)| \leq \bar{w} := \frac{2\bar{r}}{1-\gamma}$ also holds for all $t$ almost surely.*

Now we come back to the proof of Theorem 3.2. Recall that we define $F$ as the Bellman Policy Operator and the noise sequence $w(t)$ as

$$w(t) = r_t + \gamma\theta_{h(i_{t+1})}(t) - \mathbb{E}_{i' \sim \mathbb{P}(\cdot|i_t)}\left[r(i_t, i') + \gamma\theta_{h(i')}(t)\right].$$

Let $\theta^*$ be the unique solution of the equation

$$\Pi F(\Phi\theta^*) = \theta^*.$$

By the triangle inequality, we have that

$$\|\Phi \cdot \theta(T) - V^*\|_\infty \leq \|\Phi \cdot \theta(T) - \Phi \cdot \theta^*\|_\infty + \|\Phi \cdot \theta^* - V^*\|_\infty$$
$$\leq \|\theta(T) - \theta^*\|_\infty + \|\Phi \cdot \theta^* - V^*\|_\infty. \tag{61}$$

We first bound the first term of (61) by Theorem 3.1. To do this, we first rewrite the update rule of TD learning with state aggregation (6) in the form of the SA update rule (3):

$$\theta_{h(i_t)}(t+1) = \theta_{h(i_t)}(t) + \alpha_t\left(F_{i_t}(\Phi\theta(t)) - \theta_{h(i_t)}(t) + w(t)\right),$$
$$\theta_j(t+1) = \theta_j(t) \text{ for } j \neq h(i_t), j \in \mathcal{M}.$$

Now we verify all the assumptions of Theorem 3.1. Assumption 3.1 is assumed to be satisfied in the body of Theorem 3.2. As for Assumption 3.2, $F$ is $\gamma$-contraction in the infinity norm because it is the

Bellman operator, and we can set $C = \frac{2\bar{r}}{1-\gamma}$ so that $C \geq (1+\gamma)\|y^*\|_\infty$ (see the discussion below Assumption 3.2). As for Assumption 3.3, by the definition of noise sequence $w(t)$, we see that

$$
\begin{aligned}
\mathbb{E}[w(t) \mid \mathcal{F}_t] &= \mathbb{E}\big[r_t + \gamma\theta_{h(i_{t+1})}(t) - \mathbb{E}_{i' \sim \mathbb{P}(\cdot|i_t)}\big[r(i_t, i') + \gamma\theta_{h(i')}(t)\big] \mid \mathcal{F}_t\big] \\
&= \mathbb{E}\big[r_t + \gamma\theta_{h(i_{t+1})}(t) \mid \mathcal{F}_t\big] - \mathbb{E}_{i' \sim \mathbb{P}(\cdot|i_t)}\big[r(i_t, i') + \gamma\theta_{h(i')}(t)\big] \\
&= 0.
\end{aligned}
$$

In addition, we can set $\bar{w} = \frac{2\bar{r}}{1-\gamma}$ according to Proposition D.1. Finally, we can set $\bar{\theta} = \frac{\bar{r}}{1-\gamma}$ according to Proposition D.1.

Therefore, by Theorem 3.1, we see that

$$
\|\theta(T) - \theta^*\|_\infty \leq \frac{C_a}{\sqrt{T + t_0}} + \frac{C_a'}{T + t_0}, \quad \text{where} \tag{62}
$$

$$
\begin{aligned}
C_a &= \frac{40H\bar{r}}{(1-\gamma)^2}\sqrt{K_2 \log T} \cdot \sqrt{\log T + \log\log T + \log\left(\frac{4mK_2}{\delta}\right)}, \\
C_a' &= \frac{8\bar{r}}{(1-\gamma)^2}\max\left(\frac{144K_2H\log T}{\sigma'} + 4K_1(1 + 2K_2 + 4H), 2K_2\log T + t_0\right).
\end{aligned}
$$

As for the second term of (61), by Theorem D.2, we have that

$$
\|\Phi \cdot \theta^* - V^*\|_\infty \leq \frac{\zeta}{1-\gamma}. \tag{63}
$$

Substituting (62) and (63) into (61) finishes the proof.

### D.3   Proof of Proposition D.1

We show $\|\theta(t)\|_\infty \leq \frac{\bar{r}}{1-\gamma}$ by induction on $t$. The statement holds for $t = 0$ because we initialize $\theta(0) = \mathbf{0}$. Suppose the statement holds for $t$. By the induction assumption, we see that

$$
\begin{aligned}
\theta_{h(i_t)}(t+1) &= (1 - \alpha_t)\theta_{h(i_t)}(t) + \alpha_t\big[r_t + \gamma\theta_{h(i_{t+1})}(t)\big] \\
&\leq (1 - \alpha_t)\|\theta(t)\|_\infty + \alpha_t[r_t + \gamma\|\theta(t)\|_\infty] \\
&\leq (1 - \alpha_t)\frac{\bar{r}}{1-\gamma} + \alpha_t\left[r_t + \gamma \cdot \frac{\bar{r}}{1-\gamma}\right] \\
&\leq \frac{\bar{r}}{1-\gamma}.
\end{aligned}
$$

For $j \neq h(i_t), j \in \mathcal{M}$, we have that

$$
\theta_j(t+1) = \theta_j(t) \leq \|\theta(t)\|_\infty \leq \frac{\bar{r}}{1-\gamma}.
$$

Hence the statement also holds for $t + 1$. Therefore, we have showed $\|\theta(t)\|_\infty \leq \frac{\bar{r}}{1-\gamma}$ by induction.

By Theorem D.1, we know $\theta^* = \lim_{t\to\infty}\theta(t)$. Since we have already shown that $\|\theta(t)\|_\infty \leq \frac{\bar{r}}{1-\gamma}$ holds for all $t$, we must have $\|\theta^*\|_\infty \leq \frac{\bar{r}}{1-\gamma}$.

Using $\|\theta(t)\|_\infty \leq \frac{\bar{r}}{1-\gamma}$, we see that

$$
\begin{aligned}
|w(t)| &\leq |r_t| + \gamma\big|\theta_{h(i_{t+1})}(t)\big| - \big|\mathbb{E}_{i' \sim \mathbb{P}(\cdot|i_t)}\big[r(i_t, i') + \gamma\theta_{h(i')}(t)\big]\big| \\
&\leq 2\bar{r} + 2\gamma\bar{\theta} \\
&= \frac{2\bar{r}}{1-\gamma}.
\end{aligned}
$$

## D.4 Application of the SA Scheme to Q-learning with State and Action Aggregation

We study $Q$-learning with state and action aggregation in a setting that is a generalization of the tabular setting studied in [37]. Specifically, we consider an MDP $M$ with a finite state space $\mathcal{S}$ and finite action space $\mathcal{A}$. Suppose the transition probability is given by $\mathbb{P}(s_{t+1} = s' \mid s_t = s, a_t = a) = \mathbb{P}(s' \mid s, a)$, and the stage reward at time step $t$ is a random variable $r_t$ with its expectation given by $R_{s_t,a_t}$. Under a stochastic policy $\pi$, the $Q$ function (vector) $Q^\pi \in \mathbb{R}^{\mathcal{S} \times \mathcal{A}}$ is defined as

$$Q^\pi_{s,a} = \mathbb{E}_\pi \left[ \sum_{t=0}^\infty \gamma^t r_t \Big| (s_0, a_0) = (s, a) \right],$$

where $0 \leq \gamma < 1$ is the discounting factor. We use $Q^*$ to denote the $Q$ function corresponding to the optimal policy $\pi^*$.

Similar to [37], we assume the trajectory $\{(s_t, a_t, r_t)\}_{t=0}^\infty$ is sampled by implementing a fixed behavioral stochastic policy $\pi$. In $Q$-learning with state and action aggregation, the state abstraction $\psi_1$ operates on the state space $\mathcal{S}$ and the action abstraction $\psi_2$ operates on action space $\mathcal{A}$. For simplicity of notation, we define the abstraction space as $\mathcal{M} = \psi_1(\mathcal{S}) \times \psi_2(\mathcal{A})$ and the abstraction operator $h : \mathcal{S} \times \mathcal{A} \to \mathcal{M}$ as $h(s, a) = (\psi_1(s), \psi_2(a))$. The update rule for $Q$-learning with state and action aggregation is then given by

$$\theta_{h(s_t,a_t)}(t+1) = (1 - \alpha_t)\theta_{h(s_t,a_t)}(t) + \alpha_t \left[ r_t + \gamma \max_{a \in \mathcal{A}} \theta_{h(s_{t+1},a)}(t) \right],$$

$$\theta_j(t+1) = \theta_j(t) \text{ for } j \neq h(s_t, a_t).$$

(64)

As a remark, some previous work considers abstraction on the state space $\mathcal{S}$ but does not compress the action space (see [21]). In contrast, our setting also compresses the action space, and when $\psi_2$ is the identity map, our setting reduces to the case with only state aggregation.

To apply the result in Section 3.1, we define function $F$ as the *Bellman Optimality Operator*, i.e.

$$F_{s,a}(Q) = R_{s,a} + \gamma \mathbb{E}_{s' \sim \mathbb{P}(\cdot|s,a)} \max_{a' \in \mathcal{A}} Q_{s',a'}.$$

It is shown in [3] that $Q^*$ is the unique fixed point of function $F$. By viewing $\mathcal{S} \times \mathcal{A}$ as $\mathcal{N}$, we can define matrix $\Phi \in \mathcal{N} \times \mathcal{M}$ as in (4). We can rewrite the update rule (64) as

$$\theta_{h(s_t,a_t)}(t+1) = \theta_{h(s_t,a_t)}(t) + \alpha_t \big[ F_{s_t,a_t}(\Phi\theta(t)) - \theta_{h(s_t,a_t)}(t) + w(t) \big],$$

$$\theta_j(t+1) = \theta_j(t) \text{ for } j \neq h(s_t, a_t),$$

where

$$w(t) = r_t + \gamma \max_{a \in \mathcal{A}} \theta_{h(s_{t+1},a)}(t) - F_{s_t,a_t}(\Phi\theta(t))$$

$$= (r_t - R_{s_t,a_t}) + \gamma \left[ \max_{a \in \mathcal{A}} \theta_{h(s_{t+1},a)}(t) - \mathbb{E}_{s' \sim \mathbb{P}(\cdot|s_t,a_t)} \max_{a' \in \mathcal{A}} \theta_{h(s',a')}(t) \right].$$

Hence we have $\mathbb{E}[w(t) \mid \mathcal{F}_t] = 0$. In order to apply Theorem 3.1, we need the following assumption on the induced Markov chain of stochastic policy $\pi$ which is standard, cf. [37].

**Assumption D.5.** *The following conditions hold:*

1. *For each time step $t$, the stage reward $r_t$ satisfies $|r_t| \leq \bar{r}$ almost surely.*

2. *Under the behavioral policy $\pi$, the induced Markov chain $(s_t, a_t)$ with state space $\mathcal{S} \times \mathcal{A}$ satisfies Assumption 3.1 with stationary distribution $d$ and parameters $\sigma', K_1, K_2$.*

The next assumption is approximate $Q^*$-irrelevant abstraction, which measures the quality of the abstraction map and is standard in the literature (see [21]).

**Assumption D.6.** *There exists an abstract $Q$ function $q : \mathcal{M} \to \mathbb{R}$ such that $\|\Phi q - Q^*\|_\infty \leq \epsilon_{Q^*}$.*

We can now state our theorem for $Q$-learning with state aggregation.

**Theorem D.6.** *Under Assumption D.5 and D.6, suppose the step size of Q-learning with state aggregation is given by $\alpha_t = \frac{H}{t+t_0}$, where $t_0 = \max(4H, 2K_2 \log T)$ and $H \geq \frac{2}{\sigma'(1-\gamma)}$. Then, with probability at least $1 - \delta$,*

$$\|\Phi \cdot \theta(T) - Q^*\|_\infty \leq \frac{C_a}{\sqrt{T + t_0}} + \frac{C_a'}{T + t_0} + \frac{2\epsilon_{Q^*}}{1 - \gamma}, \text{ where}$$

$$C_a = \frac{40H\bar{r}}{(1-\gamma)^2} \sqrt{K_2 \log T} \cdot \sqrt{\log T + \log \log T + \log\left(\frac{4mK_2}{\delta}\right)},$$

$$C_a' = \frac{8\bar{r}}{(1-\gamma)^2} \max\left(\frac{144K_2 H \log T}{\sigma'} + 4K_1(1 + 2K_2 + 4H), 2K_2 \log T + t_0\right).$$

*Proof of Theorem D.6.* Define $\theta^*$ as the unique solution of equation $\theta = \Pi F(\Phi\theta)$, where the definition of $\Pi$ is given in (5). Under Assumption D.5, we see that $\|\theta^*\|_\infty \leq \frac{\bar{r}}{1-\gamma}$: otherwise, by assuming that $|\theta_i^*| = \|\theta^*\|_\infty > \frac{\bar{r}}{1-\gamma}$, we can derive a contradiction that $\|\Pi F(\Phi\theta^*)\|_\infty < |\theta_i^*|$. To see this, recall that linear operators $\Pi$ and $\Phi$ are non-expansions in the infinity norm (see Appendix C.1), and $\|F(v)\|_\infty < \|v\|_\infty$ for a vector $v \in \mathbb{R}^{\mathcal{N}}$ if $\|v\|_\infty > \frac{\bar{r}}{1-\gamma}$.

Further, using a similar approach with the proof of Proposition D.1, we also see that

$$\|\theta(t)\|_\infty \leq \bar{\theta} := \frac{\bar{r}}{1 - \gamma}, |w(t)| \leq \bar{w} := \frac{2\bar{r}}{1 - \gamma}$$

hold for all $t$ almost surely.

Therefore, by Theorem 3.1, we obtain that

$$\|\theta(T) - \theta^*\|_\infty \leq \frac{C_a}{\sqrt{T + t_0}} + \frac{C_a'}{T + t_0}. \tag{65}$$

To finish the proof of Theorem D.6, we only need to show that

$$\|\Phi\theta^* - Q^*\| \leq \frac{2\epsilon_{Q^*}}{1 - \gamma}. \tag{66}$$

Given the behavioral policy $\pi$, we use $\{d_{s,a} \mid (s,a) \in \mathcal{S} \times \mathcal{A}\}$ to denote the stationary distribution under policy $\pi$. Recall that we define $\mathcal{M} = \psi_1(\mathcal{S}) \times \psi_2(\mathcal{A})$. For each abstract state-action pair $(x,y) \in \mathcal{M}$, we define a distribution $p_{(x,y)}$ over $h^{-1}(x,y)$ such that

$$p_{(x,y)}(s,a) = \frac{d_{s,a}}{\sum_{(\tilde{s},\tilde{a}) \in h^{-1}(x,y)} d_{\tilde{s},\tilde{a}}}, \forall (s,a) \in h^{-1}(x,y).$$

Using the set of distributions $\{p_{(x,y)} \mid (x,y) \in \mathcal{M}\}$, we define two new MDPs:

$$M_\psi = (\psi_1(\mathcal{S}), \psi_2(\mathcal{A}), P_\psi, R_\psi, \gamma), \tag{67}$$

where $(R_\psi)_{x,y} = \mathbb{E}_{(s,a) \sim p_{(x,y)}}[R_{s,a}]$, and $P_\psi(x' \mid x, y) = \mathbb{E}_{(s,a) \sim p_{(x,y)}}[P(x' \mid s, a)]$; and

$$M_\psi' = (\mathcal{S}, \mathcal{A}, P_\psi', R_\psi', \gamma), \tag{68}$$

where $(R_\psi')_{s,a} = \mathbb{E}_{(\tilde{s},\tilde{a}) \sim p_{h(s,a)}}[R_{\tilde{s},\tilde{a}}]$, $P_\psi'(s' \mid s, a) = \mathbb{E}_{(\tilde{s},\tilde{a}) \sim p_{h(s,a)}}[P(s' \mid \tilde{s}, \tilde{a})]$.

We use $\Gamma$ to denote the Bellman Optimality Operator. For simplicity, we use the subscript to distinguish the value functions ($V^*$), the state-action value functions ($Q^*$), and the Bellman Optimality Operators ($\Gamma$) of the three MDPs $M, M_\psi$ and $M_\psi'$. Notice that $\Gamma_M$ is identical with $F$.

We can show that $\theta^*$ is identical with the state-action value function of $M_\psi$, i.e.,

$$\theta^* = Q^*_{M_\psi}. \tag{69}$$

To see this, we notice that $(\Phi\theta^*)_{s,a} = \theta^*_{h(s,a)}$. Hence we get that

$$F(\Phi\theta^*)_{s,a} = [\Gamma_M \Phi\theta^*]_{s,a}$$

$$= R_{s,a} + \mathbb{E}_{s' \sim P(s,a)}\left[\max_a (\Phi\theta^*)_{s',a}\right]$$

$$= R_{s,a} + \mathbb{E}_{s' \sim P(s,a)}\left[\max_a \theta^*_{h(s',a)}\right].$$

Using this, we further obtain that

$$
\begin{aligned}
(\Pi F(\Phi\theta^*))_{x,y} &= \sum_{(s,a)\in h^{-1}(x,y)} \frac{d_{s,a}}{\sum_{(\tilde{s},\tilde{a})\in h^{-1}(x,y)} d_{\tilde{s},\tilde{a}}} \left( R_{s,a} + \mathbb{E}_{s'\sim P(s,a)}\left[\max_a \theta^*_{h(s',a)}\right]\right) \\
&= \sum_{(s,a)\in h^{-1}(x,y)} p_{(x,y)}(s,a)\left( R_{s,a} + \mathbb{E}_{s'\sim P(s,a)}\left[\max_a \theta^*_{h(s',a)}\right]\right) \\
&= (R_\psi)_{x,y} + \sum_{(s,a)\in h^{-1}(x,y)} p_{(x,y)}(s,a) \sum_{x'\in\psi_1(\mathcal{S})} P(x'\mid s,a)\max_a \theta^*_{x',\psi_2(a)} \\
&= (R_\psi)_{x,y} + \sum_{x'\in\psi_1(\mathcal{S})} P_\psi(x'\mid x,y)\max_{y'}\theta^*_{x',y'} \\
&= [\Gamma_{M_\psi}\theta^*]_{x,y}.
\end{aligned}
$$

Since we have $\Pi F(\Phi\theta^*) = \theta^*$ by definition, we see that

$$
[\Gamma_{M_\psi}\theta^*]_{x,y} = \theta^*_{x,y}, \forall (x,y)\in\mathcal{M}.
$$

Thus we have shown that $\theta^* = Q^*_{M_\psi}$.

Next, we observe that the state-value function of MDP $M'_\psi$ is given by

$$
Q^*_{M'_\psi} = \Phi Q^*_{M_\psi}. \tag{70}
$$

This is because

$$
\begin{aligned}
\left(\Gamma_{M'_\psi}(\Phi Q^*_{M_\psi})\right)_{s,a} &= (R'_\psi)_{s,a} + \gamma\sum_{s'\in\mathcal{S}} P'_\psi(s'\mid s,a)\max_{a'}(\Phi Q^*_{M_\psi})_{s',a'} \\
&= (R'_\psi)_{s,a} + \gamma\langle P'_\psi(s,a), \Phi V^*_{M_\psi}\rangle \\
&= \sum_{(\tilde{s},\tilde{a})\in h^{-1}(h(s,a))} p_{h(s,a)}(\tilde{s},\tilde{a})\left( R_{\tilde{s},\tilde{a}} + \gamma\langle P(\tilde{s},\tilde{a}), \Phi V^*_{M_\psi}\rangle\right) \tag{71a} \\
&= \sum_{(\tilde{s},\tilde{a})\in h^{-1}(h(s,a))} p_{h(s,a)}(\tilde{s},\tilde{a}) R_{\tilde{s},\tilde{a}} \\
&\quad + \sum_{(\tilde{s},\tilde{a})\in h^{-1}(h(s,a))} p_{h(s,a)}(\tilde{s},\tilde{a})\gamma\langle P(\tilde{s},\tilde{a}), \Phi V^*_{M_\psi}\rangle \\
&= (R_\psi)_{h(s,a)} + \gamma\langle P_\psi(h(s,a)), V^*_{M_\psi}\rangle \tag{71b} \\
&= (Q^*_{M_\psi})_{h(s,a)} \\
&= (\Phi Q^*_{M_\psi})_{s,a},
\end{aligned}
$$

where we use the definition of $M'_\psi$ (see (68)) in (71a); we use the definition of $M_\psi$ (see (67)) in (71b).

By (70), we see that

$$
\left\|\Phi Q^*_{M_\psi} - Q^*_M\right\|_\infty = \left\|Q^*_{M'_\psi} - Q^*_M\right\|_\infty \le \frac{1}{1-\gamma}\left\|\Gamma_{M'_\psi} Q^*_M - Q^*_M\right\|_\infty. \tag{72}
$$

We further notice that

$$\left|(\Gamma_{M_\psi}^* Q_M^*)_{s,a} - (Q_M^*)_{s,a}\right|$$

$$= \left|(R_\psi')_{s,a} + \gamma\langle P_\psi(s,a), V_M^*\rangle - (Q_M^*)_{s,a}\right|$$

$$= \left|\left(\sum_{(\tilde{s},\tilde{a})\in h^{-1}(h(s,a))} p_{h(s,a)}(\tilde{s},\tilde{a})(R_{\tilde{s},\tilde{a}} + \gamma\langle P(\tilde{s},\tilde{a}), V_M^*\rangle)\right) - (Q_M^*)_{s,a}\right| \tag{73a}$$

$$= \left|\sum_{(\tilde{s},\tilde{a})\in h^{-1}(h(s,a))} p_{h(s,a)}(\tilde{s},\tilde{a})((Q_M^*)_{\tilde{s},\tilde{a}} - (Q_M^*)_{s,a})\right|$$

$$\leq \sum_{(\tilde{s},\tilde{a})\in h^{-1}(h(s,a))} p_{h(s,a)}(\tilde{s},\tilde{a})|(Q_M^*)_{\tilde{s},\tilde{a}} - (Q_M^*)_{s,a}|$$

$$\leq \sum_{(\tilde{s},\tilde{a})\in h^{-1}(h(s,a))} p_{h(s,a)}(\tilde{s},\tilde{a})(2\epsilon_{Q^*}) \tag{73b}$$

$$= 2\epsilon_{Q^*},$$

where we use the definition of $M_\psi$ in (73a); we use Assumption D.6 in (73b).

Substituting (73) into (72) gives that

$$\left\|\Phi Q_{M_\psi}^* - Q_M^*\right\|_\infty \leq \frac{2\epsilon_{Q^*}}{1-\gamma}. \tag{74}$$

Combining (69) and (74) finishes the proof. $\qquad\square$