# OpenReview forum: "Multi-Agent Reinforcement Learning in Stochastic Networked Systems"
_NeurIPS.cc/2021/Conference — NeurIPS 2021 Poster_

### Official Review · Reviewer_fBbm · 2021-07-14

**Rating:** 7
**Confidence:** 4

**Summary:**

In this paper, the authors propose a scalable actor-critic framework for a class of multi-agent reinforcement learning (MARL) problems with a stochastic and non-local network of agents. The authors provide a finite-time error bound for the algorithm and show the dependency of the convergence rate on the speed of information spread in the network.

**Main Review:**

Pros:

MARL with network structure is an important and challenging problem. Stochastic and nonlocal networks are even harder problems to tackle under the MARL framework. The theoretical contributions of this paper are solid and the techniques used are different from the work with deterministic networks [a].

[a] G. Qu, Y. Lin, A. Wierman, and N. Li. Scalable multi-agent reinforcement learning for networked systems with average reward. Advances in Neural Information Processing Systems, 33, 2020.

Comments:

(1) Is it possible to include a neural-network-type-of functional approximation (see [b] and [c])? This will make the framework more general and applicable to broader problems.
[b] Lingxiao Wang, Qi Cai, Zhuoran Yang, and Zhaoran Wang. Neural policy gradient methods: Global optimality and rates of convergence.arXiv preprint arXiv:1909.01150, 2019.
[c] Qi Cai, Zhuoran Yang, Jason D Lee, and Zhaoran Wang. Neural temporal-difference learning converges to global optima.Advances in Neural Information Processing Systems, 32, 2019.


(2) I understand the importance of the preliminary analysis in Section 2 (for stochastic approximation) and Section 3 (for state aggregation). However, the main framework of networked MARL is not introduced until page 5. The authors may consider restructuring the materials a bit.


=============After authors' response=============
Thanks for the response. I read the comments from other reviewers and the authors' responses to all reviewers. RL with stochastic networked systems is a hard problem and I think this paper has a solid contribution to the theoretical front of MARL. My only concern was the organization of the original submission and the authors agreed to rearrange the materials, which I believe is quite achievable for the camera-ready version.

Therefore I will maintain my original score of 7.

**Time Spent Reviewing:**

5

---

> ### Author Response · Authors · 2021-08-10
> **Response to Reviewer fBbm**
>
> Thank you for your insightful comments. Below are the responses to your specific comments.
>
> 1. (Function approximation) Incorporating more general types of function approximation methods as you mentioned will certainly make the algorithm more applicable. A direction we are working on is to first do a $\kappa$-hop state aggregation to reduce $(s, a)$ to $\left(s_{N_i^\kappa}, a_{N_i^\kappa}\right)$, and then use a local neural network to approximate the local Q-function of ${\left(s_{N_i^\kappa}, a_{N_i^\kappa}\right)}$. While the first step of $\kappa$-hop state aggregation leverages the networked interaction structure to reduce the size of state/action space, the second step of neural function approximation will be particularly useful when the $\kappa$-hop state/action is still very large. The major challenge is on deriving a corresponding generalization of Thm 3.1 about TD approximation error for this new composed function approximation method. We will add this discussion in the revision.
>
> 2. (Organization) Yes, we agree the current manuscript introduces MARL too late. In our revision, we will reorder the sections to place the networked MARL section before stochastic approximation and TD learning with state aggregation. We also plan to shorten the later two sections to give more space to the main part of networked MARL.

---

### Official Review · Reviewer_QTUD · 2021-07-16

**Rating:** 6
**Confidence:** 4

**Summary:**

This paper first study the finite time analysis of linear SA with Markov noise, then apply the result to study TD with state aggregation. This paper further study the convegence of MARL based on exp-decay assumption.

The result is not strong and the author did not make sufficient comparison with the state-of-the-art. Specifically, the rate of linear SA/TD is not compared with the state-of-the-art in terms of conditional number dependence. The rate of AC algorithm seems to be very slow but the author did not make justification for that.

**Ethics Review Area:**

["I don’t know"]

**Limitations And Societal Impact:**

This paper does not have potential negative societal impact.

**Main Review:**

(1) The linear SA with “aggregated entries" is very similar to TD with linear function approximation setting, which has be extensitvely studied before. The author did not make it clear the technical novelity compare with previous proof.

(2) In (4), the author implicitly assume that $\Phi^\top D \Phi$ is invertable. This assumption is fine for linear TD with independent features, but seems to be strong in the setting considered in this paper. The author should at least provide some justifications here.

(3) It is not clear whether the author obtain a tighter bound for TD learning compare with the state-of-art [1].

(4) Although it is a theorecial paper, empirical verification is also needed becasue this paper proposes a new algorithm here.



[1] Kotsalis, G., Lan, G., & Li, T. (2020). Simple and optimal methods for stochastic variational inequalities, II: Markovian noise and policy evaluation in reinforcement learning. arXiv preprint arXiv:2011.08434.

**Time Spent Reviewing:**

4h

---

> ### Author Response · Authors · 2021-08-10
> **Response to Reviewer QTUD**
>
> Thank you for your insightful comments.
>
> Given that your review focused on the results relating to Stochastic Approximation (SA) and TD learning with state aggregations, we want to emphasize that the primary goal of this paper is to contribute to the theory of multi-agent reinforcement learning by providing, to the best of our knowledge, the first finite-time error bounds for MARL in settings where agents interact according to a time-varying network.  Highlights of our result include proving a $\mu$-decay result (Thm 4.3) for the MARL setting under stochastic networks, which sheds light on the structure of the exponentially-large $Q$-function of the MARL problem. Such a $\mu$-decay result generalizes the exponential decay results known in the literature (ref. [36, 38] in the paper) for MARL with static local interaction structure. Proving such a result is non-trivial, as the agents no longer interact locally (each agent has a non-zero probability of interacting with any other agent), and requires novel proof techniques compared to earlier works (ref. [36, 38] in the paper) that have relied on the static local interaction structure.
>
> In proving the MARL results, we also develop the results on SA and TD learning with state aggregations because they are key components in the proof and can provide key intuition on why the distributed algorithm works. We chose to present the results in SA and TD learning because we believe they are of independent interest (e.g., we apply the SA results to obtain novel finite-time error bounds for Q-learning with state-aggregation in Appendix C.4.). To prevent future confusion, in our revision, we will make our contribution more clear by reordering the sections and presenting the MARL part first.  We hope this will help avoid confusion about our contribution.
>
> Here is the response to your specific comments.
>
> 1. Our SA with state aggregation (2) is not linear because we only require the operator $F$ to be a contraction (Assump 2.2), which makes it different from the linear SA considered in [43]. Although the generality on nonlinearity is not necessary for TD learning (see line 210), it is of independent interest due to its application to Q-learning with state-aggregation (see Appendix C.4), where $F$ is no longer linear (see line 949). Further, as we discussed in lines 222-226, the TD convergence result we derived in Thm 3.1 is also different from previous works [43, 4] as our result uses the infinity norm and does not require the projection step. Besides, when compared with [43], our result is a concentration bound (bound on error with high probability) while [43] is a bound on the expectation of the squared error, and the two do not directly imply each other. That being said, we view our results on TD learning as complementary to these earlier works, and our results are more suitable for state aggregation and analyzing the networked MARL convergence in later sections.
>
> 2. Our setting of TD learning with state aggregation is a special case of linear TD with independent features. To see this, note that the feature vector associated with each aggregated state only consists of zeros and ones, and any two feature vectors cannot have nonzero entries at the same index. In networked MARL, $\Phi^T D \Phi$ is a diagonal matrix whose entries are the probabilities of visiting each $\kappa$-hop state-action pair under the stationary distribution ($d^\theta(z’)$ defined in Assumption 4.1). It is reasonable to assume such probabilities are lower bounded by some positive constants when $\kappa$ is relatively small compared with the diameter of the underlying graph. We will add this discussion.
>
> 3. The results in [Kotsalis et al., 2021] cannot be used to derive a bound in the form of Thm 3.1 for the TD learning algorithm we considered. To see this, consider the tabular TD learning, which is a special case of TD learning with state aggregation (Sec 3) where $h$ is an identity mapping. In this case, our TD learning algorithm (update rule (5)) corresponds to the case $V(x_t, x) = \frac{1}{2}\left\lVert{x - x_t}\right\rVert_2^2$ in [Kotsalis et al., 2021]. Applying Corollary 3.5 in [Kotsalis et al., 2021] directly will give a finite-time error bound on the value function, where the left hand side is $\left\lVert{\theta(T) - V^*}\right\rVert_2^2$ and the right hand side contains $\left\lVert{\theta(0) - V^*}\right\rVert_2^2$. This is different with Thm 3.1 where the left hand is $\left\lVert{\theta(T) - V^*}\right\rVert_\infty$. On the other hand, we agree that the results in [Kotsalis et al., 2021] can potentially be used to design a different TD learning algorithm that has better theoretical guarantees than the current one. To achieve this, we may need to design the function $V$ in [Kotsalis et al., 2021] in order to obtain an error-bound in the infinity norm, and we may also incorporate techniques like CTD and FTD.
>
> 4. We did some simulations on two specific instances of the application examples given in Appendix A.1 and A.2. For the wireless network example in Appendix A.1, we experiment on the setup shown in Figure 1, where 24 user nodes are distributed in a $3 \times 4$ grid and each grid contains 2 users. For the spreading network example in Appendix A.2, we experiment on the setup shown in Figure 2, where 25 agents are located in a $5 \times 5$ grid network and the cost/transition parameters are sampled randomly. Note that the spreading network example is beyond the scope of previous works [36, 38] since it has a non-static interaction structure. Both simulations show that, starting from an initial policy that picks each local action uniformly randomly, our MARL algorithm can make significant improvements through learning and outperform a static benchmark based on the local ALOHA protocol in the wireless network example. Due to space limits and the theoretic nature of our work, we didn’t include these simulation results in the initial submission. We are happy to include these results in revision.

---

> > ### Comment · Reviewer_QTUD · 2021-08-31
> > **Thanks for response**
> >
> > The author's response has address most of my concerns. I will increase my score accordingly.

---

### Official Review · Reviewer_pJBQ · 2021-07-17

**Rating:** 6
**Confidence:** 2

**Summary:**

The paper proposes a Scalable Actor-Critic algorithm for a stochastic network of agents. First of all, the authors introduce a new finite-time analysis of stochastic approximation. Then they show how stochastic approximation can be used for state aggregation. Finally they proposed the main algorithms and the theoretical results.

**Limitations And Societal Impact:**

In my opinion, this paper has not potential negative societal impact.

**Main Review:**

The paper takes into account the problem of learning the optimal policies for stochastic network agents.
The problem is interesting, and the authors show by examples that real-world problems can be cast into this framework. I appreciated the effort of the authors to motivate the considered setting.

However, I have found the paper difficult to follow and hard to read. I think that the main problem is due to the space spent for the stochastic approximation part instead then the main objective of the paper, i.e., the algorithm for stochastic network agent. In fact, until page 6, the network setting is still not presented, and so the reader does not understand what the goal of the paper is. Moreover, the main algorithm is presented only on page 8 and this leads to having only one page to discuss it. I would like to strongly suggest removing the part on state aggregation to give more space to the main part of the paper.

Going into details, the main difference between the setting proposed in this paper and the one of Du et al. is that the transitions and the rewards do not depend on static links between the agent, but that the links are sampled by a distribution $D$. An interesting aspect that is not enough discussed in the paper is: what happens to the proposed algorithm if it is in the setting considered by Du et al? Does it achieve better theoretical performance?

I have found particularly nice the fact that the $\mu$ decay property enjoys small-world phenomena.

I think that this paper needs an experimental evaluation to compare the proposed approach with the one of Du et al. when it is in the same setting of Du et al. (since the considered setting is a generalization of the one of Du et al.). I will suggest following the same line of experiments as in Du et al. and constructing similar scenarios also for the generalized setting considered in this paper.

I will also suggest adding a conclusive section to underline the main results of this paper and to highlight the differences with the state-of-the-art works.


Post author response:
I would like to thank the authors for their detailed answers. I accordingly increase my score (5 --> 6). I am confident that the authors will follow my suggestion to move the experimental evaluation to the main paper and to add a conclusive section.

**Time Spent Reviewing:**

4

---

> ### Author Response · Authors · 2021-08-10
> **Response to Reviewer pJBQ**
>
> Thank you for your insightful comments. One thing we are not sure about is the reference “Du et. al.” you mentioned. We haven’t cited any paper “Du et. al.” in our reference list, and based on our reading of your comments, we believe by “Du et. al.” you actually meant “Qu et. al.” ([38] in our references). Please correct us if our understanding is incorrect.
>
> Here is our response to your detailed comments.
>
> 1. (Organization) In our revision, we will reorder the sections to place the networked MARL section before stochastic approximation and TD learning with state aggregation. We also plan to shorten the later two sections to give more space to the main part of networked MARL.
>
> 2. (Algorithm) Our algorithm is a generalization of the one in [38] as we discussed in lines 337-340. In the more restricted setting of [38], if we set $\beta = 0$ in our algorithm, then our algorithm recovers the one in [38]. In terms of our analysis, we show a stronger bound (see lines 371-373) than [38] under a different assumption to ensure sufficient exploration (Assumption 4.1).  This is a contribution in the deterministic setting in addition to the generalization to the stochastic setting. Lastly, we emphasize that despite the connection to [38] in terms of the algorithm, one of the major difficulties in the stochastic network setting is to prove the $\mu$-decay property, for which the techniques and results in [38] are far from sufficient, and so we have developed novel proof techniques to show the decay property in our setting.
>
> 3. (Small World Phenomena) We want to point out that, as we discussed in lines 324-328, the $\mu$-decay property deteriorates when the small-world phenomena becomes stronger. Intuitively, this suggests that localized learning algorithms prefer environments where information propagates ‘slowly’, which is to be expected. As we mentioned in line 327, an interesting future direction is to show a weaker $\mu$-decay property still holds even if small-world phenomena do occur. We are very excited to further explore this connection.
>
> 4. (Experiment and conclusion) We are happy to include experiment results and a conclusion section in revision. For the experiments, we demonstrated the efficacy of our algorithm on two specific instances of the application examples given in Appendix A.1 and A.2. The second experiment on the spreading network involves non-static interaction graphs, for which the results in [38] cannot be applied. The first wireless network example only contains static interaction graphs and thus can also fit in the settings of [36, 38] when each agent’s policy is purely local (i.e., $\beta = 0$). In contrast with [36, 38], our algorithm allows each agent’s policy to not just depend on its own local state, but also other agents’ states (i.e., $\beta>0$). Therefore, given the enlarged policy dependence, we expect our algorithm to perform better than [38] in this example.  Due to space limits and the theoretic nature of our work, we didn’t include these simulation results in the initial submission. We are happy to include them in revision.
>
> Thank you again for your detailed and insightful comments.

---

### Official Review · Reviewer_savS · 2021-08-02

**Rating:** 7
**Confidence:** 3

**Summary:**

The paper addresses a class of MARL problems where a stochastic network of agents interacts cooperatively. The give a scalable actor-critic algorithm with finite-time convergence guarantees. The guarantees are obtained via a general stochastic approximation scheme that incorporates state aggregation.

**Limitations And Societal Impact:**

-

**Main Review:**

The paper is well written and overall I think it is well organized, though one could argue that Section 4 should be presented a bit earlier, since it gives much of the setting for the paper. However, I don't feel strongly either way, and I am fine with starting from the abstract/general techniques in Section 2. If the paper gets accepted, I recommend fleshing out the description of the algorithm if the conference allows additional space for the final version.

I think the authors do a good job at motivating the new setting. I very much appreciated the results in Section 4, I think they are articulated very clearly and at the right pace. The SA scheme given in Section 2 seems fairly general, though I should note that I am not an expert on this specific sub-area, so it's hard for me to evaluate whether all relevant literature has been appropriately cited and how much incremental the results are.

All in all, I found the conceptual setup convincing, the paper well written and the results of interest, at least theoretically. To me, the biggest weakness of the paper is its lack of experimental evaluation, which seems standard and expected at this venue. In particular, I think that the paper would be significantly stronger if the authors could also show positive experimental results, possibly comparing to the related work by Qu et al [38]---the latter provides experimental data in their Section 4. However, I think the paper brings enough to the table that I am happy to suggest acceptance nonetheless.

**Time Spent Reviewing:**

5

---

> ### Author Response · Authors · 2021-08-10
> **Response to Reviewer savS**
>
> We would like to thank the reviewer for the insightful comments.
>
> We have done simulations for the two specific instances of the application examples given in Appendix A.1 and A.2. For the wireless network example in Appendix A.1, we experiment on the setup shown in Figure 1, where 24 user nodes are distributed in a $3 \times 4$ grid and each grid contains 2 users. For the spreading network example in Appendix A.2, we experiment on the setup shown in Figure 2, where 25 agents are located in a $5 \times 5$ grid network and the cost/transition parameters are sampled randomly. Note that the spreading network example is beyond the scope of previous works [36, 38] since it has a non-static interaction structure. Both simulations show that, starting from an initial policy that picks each local action uniformly randomly, our MARL algorithm can make significant improvements through learning and outperform a static benchmark based on the local ALOHA protocol in the wireless network example. Due to space limits and the theoretic nature of our work, we didn’t add these simulation results in the initial submission. We are happy to include these results in revision.

---

### Decision · Program_Chairs · 2021-09-27

**Decision:**

Accept (Poster)

**Comment:**

While the initial reviews were contrasting, the rebuttals helped the reviewers to appreciate better the work. The major issue raised by the Reviewers concerns the organization of the work. All the Reviewers feel that the paper should be re-organized as the authors discuss in their rebuttals. I invite the authors to do that in the camera ready. A further, but secondary, issue concerns the lack of empirical results. My opinion is that this are not strictly necessary as the paper is mainly theoretical. However, having them in the paper would make it stronger. So, I invite the authors to add some empirical results in the final version of the paper.